# AWM: ACCURATE WEIGHT-MATRIX FINGERPRINT FOR LARGE LANGUAGE MODELS

**Boyi Zeng**[1*]**, Lin Chen**[1,4*]**, Ziwei He**[2]**, Xinbing Wang**[3]**, Zhouhan Lin**[1,2†]

[1]LUMIA Lab, School of Artificial Intelligence, Shanghai Jiao Tong University,
[2]Shanghai Innovation Institute, [3]Shanghai Jiao Tong University, [4]Fudan University
boyizeng@sjtu.edu.cn, {charliecl0526, lin.zhouhan}@gmail.com

## ABSTRACT

Protecting the intellectual property of large language models (LLMs) is crucial, given the substantial resources required for their training. Consequently, there is an urgent need for both model owners and third parties to determine whether a suspect LLM is trained from scratch or derived from an existing base model. However, the intensive post-training processes that models typically undergo—such as supervised fine-tuning, extensive continued pretraining, reinforcement learning, multi-modal extension, pruning, and upcycling—pose significant challenges to reliable identification. In this work, we propose a training-free fingerprinting method based on weight matrices. We leverage the Linear Assignment Problem (LAP) and an unbiased Centered Kernel Alignment (CKA) similarity to neutralize the effects of parameter manipulations, yielding a highly robust and high-fidelity similarity metric. On a comprehensive testbed of 60 positive and 90 negative model pairs, our method demonstrates exceptional robustness against all six aforementioned post-training categories while exhibiting a near-zero risk of false positives. By achieving perfect scores on all classification metrics, our approach establishes a strong basis for reliable model lineage verification. Moreover, the entire computation completes within 30s on an NVIDIA 3090 GPU.[1]

## 1 INTRODUCTION

Large language models (LLMs) have become foundational to many artificial intelligence applications. However, training an LLM from scratch demands substantial computational resources and vast amounts of data. Consequently, most open-source models are released under specific licenses (Touvron et al., 2023a; Team et al., 2025; Mesnard et al., 2024; Kamath et al., 2025) or require an application (Touvron et al., 2023b; Zhang et al., 2022; Penedo et al., 2023; BaiChuan-Inc, 2023; Team, 2023; Zheng et al., 2023; Grattafiori et al., 2024) and approval process to protect intellectual property. Despite these measures, some developers may circumvent such protections by wrapping or post-training existing base LLMs, then falsely claiming to have trained their own models. Recent controversies (pzc163 et al., 2024; Yoon et al., 2025) have underscored the urgent need to determine whether a suspect model is genuinely trained from scratch or derived from an existing base model.

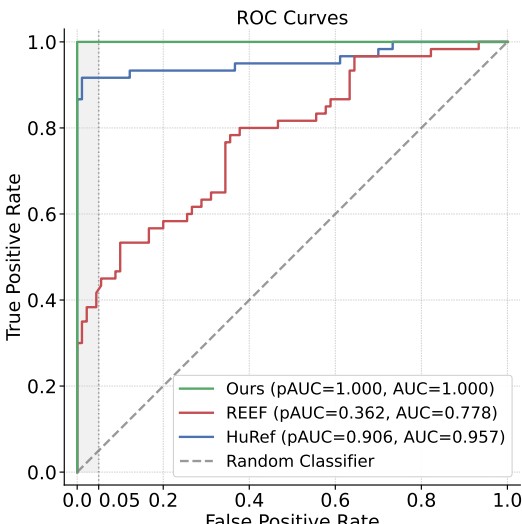

Figure 1: The ROC curve on 150 pairs LLMs. Our method perfectly distinguishes between related and unrelated LLMs (pAUC=1.0, AUC=1.0), significantly outperforming the baselines.

---

[*] Equal contribution    [†] Corresponding author
[1]The code is available at https://github.com/LUMIA-Group/AWM.

The core challenge lies in extracting a stable fingerprint to identify the true base model. This task is complicated by the fact that LLMs often undergo heavy post-training processes, such as supervised fine-tuning (SFT), extensive continued pretraining (Team et al., 2024; Hui et al., 2024), and reinforcement learning (RL). Furthermore, emerging techniques like extension to multimodal tasks, architectural pruning (Xia et al.), and upcycling (He et al., 2024) can drastically alter a model's parameters, outputs, and even its structure, posing substantial challenges for identification. Moreover, malicious actors might intentionally manipulate weight matrices through operations including scaling, permutation, pruning, and even rotation to obscure a model's origin.

Therefore, a practical fingerprinting method needs to satisfy the following critical conditions:

- **Robustness** against extensive post-training processes.
- **Resilience** to malicious weight manipulations such as scaling, pruning, permutation, and rotation.
- **Performance Preservation**, ensuring no degradation of the LLM's capabilities, as both users and manufacturers place a high premium on model performance.
- **High Fidelity**, possessing sufficient discriminative power and an extremely low false-positive rate to prevent false accusations of model theft.
- **Computational Efficiency**, remaining lightweight enough for comparisons given the immense parameter counts of modern LLMs.

A review of previous work reveals that existing methods often fail to meet one or more of these criteria. Watermarking techniques, for instance, embed identifiable signals via extra training (Peng et al., 2023; Xu et al., 2024), but this can degrade performance (Russinovich & Salem, 2024), may not survive aggressive post-training (Fernandez et al., 2024; Gubri et al., 2024), and must be applied before the model's release. Other existing fingerprinting methods either suffer from high false-positive rates (Zhang et al., 2025) or lack robustness against heavy post-training modifications like continued pretraining (Zeng et al., 2024), a limitation we confirm in our experiments.

In this work, we propose a novel approach that satisfies all the aforementioned requirements. Our method begins with an analysis of LLM weight manipulations. By leveraging the Linear Assignment Problem (LAP) and an unbiased Centered Kernel Alignment (CKA), we derive a similarity metric that is robust to all these manipulations. Our entire similarity computation completes within 30 seconds on a single NVIDIA 3090 GPU. Furthermore, our method is training-free and does not impair the LLM's performance.

We validate our approach on a comprehensive test set of 60 positive (base-offspring) and 90 negative (independent) model pairs. Compared to state-of-the-art methods such as HuRef and REEF, our approach achieves a much larger separation gap while maintaining a near-zero false-positive risk. Crucially, our method is the only one to prove robust against all tested forms of post-training: SFT, extensive continued pretraining (up to 5.5T tokens), reinforcement learning, multi-modal extension, pruning, and upcycling. On this 150-pair dataset, our method achieves a perfect Area Under the Curve (AUC), partial AUC (for False Positive Rate $< 0.05$), and True Positive Rate @ 1% False Positive Rate of 1.0, establishing a strong basis for reliable and robust model lineage verification.

## 2 RELATED WORK

Model copyright protection methods fall into two main categories: watermarking and fingerprinting.

**Watermarking.** Watermarking methods typically involve finetuning models to inject backdoor triggers that prompt the model to generate predefined content, or embedding watermarks directly into model weights for identification purposes. A substantial body of research has explored watermarking for smaller DNNs such as CNNs and BERT (Devlin et al., 2019), including encoding watermarks into model weights (Chen et al., 2019a; Wang & Kerschbaum, 2021; Liu et al., 2021; Uchida et al., 2017) and injecting triggers to produce specific outputs (Adi et al., 2018; Guo & Potkonjak, 2018; Le Merrer et al., 2020; Chen et al., 2019b). However, these methods are often task-specific and not well-suited for foundation LLMs. For watermarking LLMs, researchers have proposed various methods to inject watermarks for identification (Russinovich & Salem, 2024; Xu et al., 2024; Li et al., 2023; Peng et al., 2023; Kirchenbauer et al., 2023; Zhao et al., 2023). Nevertheless, these approaches inevitably compromise LLM performance and are not robust to extensive post-training.

**Fingerprinting.** Fingerprinting methods extract intrinsic model features as signatures for identification without requiring additional training, thereby preserving model performance. These methods are generally categorized based on the auditor's access level: white-box (full access) and black-box (API access).

*White-box Fingerprinting.* When model parameters are accessible, auditors can derive fingerprints from static weights or dynamic internal states. For small DNNs, prior works (Zhao et al., 2020; Pan et al., 2022; Yang et al., 2022; Lukas et al., 2019; Peng et al., 2022) typically analyze model behaviors on preset test cases. In the context of LLMs, HuRef (Zeng et al., 2024) is a representative static method that derives invariant terms from weight matrices to compute similarity. However, it is not robust to extensive training. Similarly, Yoon et al. (2025) utilize the standard deviation distributions of attention matrices as fingerprints; while stable under training, such statistical measures may carry a risk of false positives. Moving beyond static weights, other methods investigate dynamic signals. For instance, DeepJudge (Chen et al., 2022) and EasyDetector (Zhang et al., 2024) utilize intermediate activation values to quantify model distance. REEF (Zhang et al., 2025) further measures the geometric similarity of representation spaces but also suffers from a high false positive rate. More recently, TensorGuard (Wu et al., 2025) proposes utilizing the statistical features of gradients generated during backpropagation as a stable signature to characterize the model's optimization landscape.

*Black-box Fingerprinting.* With the proliferation of Model-as-a-Service (MaaS), black-box methods that rely solely on API input-output interactions have gained traction. These approaches generally follow two paradigms: analyzing output distributions or constructing specific trigger queries. The former focuses on identifying unique stylistic idiosyncrasies or probability distributions inherent to a model family. For example, LLMmap (Pasquini et al., 2025) and other distributional approaches (Yang & Wu, 2024; Gao et al., 2025; McGovern et al., 2025) query the model with general prompts to capture distinct response patterns or logit features. The latter paradigm involves optimizing specific "trap" inputs or adversarial queries (Xu et al., 2024; Gubri et al., 2024; Jin et al., 2024; Xu et al., 2025) designed to force a suspect model to output a predefined unique response.

However, black-box methods face significant robustness challenges. Fundamentally, relying solely on surface-level outputs results in a loss of critical information regarding the model's internal mechanisms compared to white-box access (Shao et al., 2025). Consequently, these methods are often fragile to post-training modifications such as instruction tuning or simple system prompt changes, which can disrupt the fingerprint (Xu et al., 2025; Tsai et al., 2025). To date, there is still a lack of fingerprinting methods for LLMs that are both robust to extensive training and exhibit a very low risk of false positives.

## 3 PRELIMINARY

**LLM Architecture** Most Large Language Models (LLMs) follows a decoder-only Transformer architecture (Radford et al., 2018). The details of various LLMs may differ, yet the Transformer blocks are similar. In particular, a Transformer block in an LLM usually consists of residual connections (He et al., 2016), Root Mean Square Normalization (RMSNorm, Zhang & Sennrich (2019)), the self-attention mechanism (Lin et al., 2017), Rotary Position Embedding (RoPE, Su et al. (2024)), and a feed-forward network (FFN). We denote by $\mathbb{W}_A$ the set of an $L$-layered LLM A's weights, and $\mathbb{W}_{A,\text{partial}} = \{W_{A,\text{emb}}\} \bigcup_{l=1}^{L} \{W_{A,i}^{(l)} \mid i \in \{Q, K\}\}$ the set of word embeddings and Q, K matrices, where $W_{A,\text{emb}}$ is the word embeddings and $W_{A,Q}^{(l)}, W_{A,K}^{(l)}$ are the Q, K matrices at the $l$-th layer. We defer the rest of the notations to Appendix D.1.

**Central Kernel Alignment (CKA)** Proposed in Kornblith et al. (2019), CKA is a similarity detection method (Zhang et al., 2025) based on Hilbert-Schmidt Independence Criterion (HSIC, Gretton et al. (2005)). It is invariant to column-wise orthogonal transformations and constant multiplications:

**Theorem 3.1** *For any input matrices $X_1 \in \mathbb{R}^{m \times n_1}, X_2 \in \mathbb{R}^{m \times n_2}$, any orthogonal transformations $U_1 \in \mathbb{R}^{n_1 \times n_1}, U_2 \in \mathbb{R}^{n_2 \times n_2}$, any non-zero constants $c_1, c_2 \in \mathbb{R}$,*

$$CKA(X_1, X_2) = CKA(c_1 X_1 U_1, c_2 X_2 U_2). \tag{1}$$

Even with semi-orthogonal $U_1$ and $U_2$ where $U_1 \in \mathbb{R}^{n_1 \times n_1'}, U_2 \in \mathbb{R}^{n_2 \times n_2'}, n_1 > n_1', n_2 > n_2'$, CKA still preserves input similarity to some extent (Kang et al., 2025; Chun et al., 2025). Nevertheless, the standard HSIC estimator converges at rate $1/\sqrt{m}$ and shows finite-sample bias (Gretton et al., 2005; Murphy et al., 2024). To this end, an unbiased version (Song et al., 2007) is proposed and extends the value of CKA from $[0, 1]$ to $[-1, 1]$. Additionally, linear kernels are often selected in CKA due to their computational efficiency and similar performance to other kernels (Kornblith et al., 2019). We give the formal definition of CKA in Appendix D.2, and the proof of Theorem 3.1 in Appendix E.1.

## 4 MANIPULATIONS ON AN OPEN-SOURCE LLM

In this part, we investigate which matrix-weight manipulations can remain compatible with preserving a model's behaviour. We first present the definition of matrix weight manipulation in §4.1. Then, we examine how key components of an LLM, including residual connections (§4.2), RMSNorm (§4.3), and RoPE together with attention scores (§4.4), can constrain and gradually narrow the space of admissible manipulations when the manipulated model is required to produce outputs similar to the base model. Finally, we derive potential attacks on matrix weights in §4.5 based on these constraints, with a focus on the Q, K matrices and the embedding layer.

### 4.1 PROBLEM DEFINITION

Weights of an open-source LLM are often inherited, but unclaimed inheritance invites manipulations. Although the source code may not explicitly reveal these manipulations, the weights are vulnerable to modifications that leave no trace in code (e.g., post-hoc matrix multiplications to produce new weights), and they can even be altered to evade independence tests. To enable detection on model weights, we first formalize the plausible manipulation forms.

**Definition 4.1 (LLM Weight Manipulations)** *Let A and B be two open-source LLMs with the same number of layers L, and $\mathbb{W}_A, \mathbb{W}_B$ denote the sets of matrix weights for A and B respectively. Then, if B manipulates A, the manipulations on $\mathbb{W}_{A,partial}$ are*

$$W_{B,i}^{(l)} = L_{B,i}^{(l)} W_{A,i}^{(l)} R_{B,i}^{(l)} + E_{B,i}^{(l)} \quad \text{for all } 1 \leq l \leq L \text{ and } i \in \{Q, K\} \tag{2}$$

$$W_{B,emb} = W_{A,emb} R_{B,emb} + E_{B,emb} \tag{3}$$

*where L and R are row-wise and column-wise transformation matrices, and E is the error term. The learnable weights in RMSNorm are also changed correspondingly.*

We omit the row-wise transform on word embeddings, $L_{B,\text{emb}}$, since each row of $W_{A,\text{emb}}$ represents a token and mixed token representations are hard to be faithfully recovered in the calculation of attention scores. We further assume that the suspicious and manipulated models produce similar (or identical) outputs to be consistent with the goal of reusing base model performance. This assumption induces constraints on the transformation matrices and, in turn, enables detection via weight-matrix similarity tests. We first develop a fine-grained view of the constraints on $\mathbb{W}_{A,\text{partial}}$ in what follows.

### 4.2 RESIDUAL CONNECTIONS: PASSING MANIPULATIONS FORWARD

In what follows, we first analyze how residual connections propagate weight manipulations forward through the network while preserving the model's outputs. A Transformer block consists of multiple components linked by residual connections. If one component takes input $X$, the next receives $Y = X + f(X)$, so any manipulation $T$ on $X$ must be recovered within the component to propagate as the next input (Zeng et al., 2024). This propagation also depends on the constituent functions of the component, which either commute with the manipulation to recover it or remain invariant and absorb it. We formalize this in the following:

**Proposition 4.2 (Proof in Appendix E.2)** *Let $f = f_n \circ \cdots \circ f_1$ be a component in Transformer and let T be a manipulation in the input. If for every $k \in \{1, \ldots, n\}$, $T \circ f_k = f_k \circ T$, then $f \circ T = T \circ f$, i.e. the manipulation propagates through constituent functions of a component.*

Compared to Zeng et al. (2024), Proposition 4.2 shows that residual connections are more vulnerable under a component-wise view, since constituent functions can propagate manipulations in various

ways. However, nonlinearities of the functions, especially those within the self-attention mechanism, pose constraints for manipulations on both inputs and weights. We next split the self-attention mechanism into two constituent functions, RMSNorm and attention (see Definition D.4), and show how these effects arise and are constrained these two functions with a focus on RMSNorm, RoPE and attention scores.

### 4.3 RMSNorm: A Constraint on Embedding Manipulations

Next, we examine how RMSNorm may allow certain manipulations of word embeddings. Any input to the self-attention mechanism, including word embeddings and hidden states, first go through RMSNorm. However, RMSNorm can facilitate potential manipulations on inputs, because it commutes with certain transformations $R_{B,\text{emb}}$ of the embeddings:

**Theorem 4.3** *Let models $A$ and $B$ share the same[2] architecture. Let $c \neq 0$ be a scalar, $P$ be a (partial) permutation matrix, and $D$ be a signature matrix. Then, $R_{B,emb} = cPD$ can be recovered after RMSNorm in model B if related RMSNorm parameters are adjusted.*

Theorem 4.3 indicates that RMSNorm is susceptible to embedding manipulations composed of constant multiplications, permutations and sign flips. On the other hand, other manipulations on word embeddings are generally neither commutative with RMSNorm nor invariant to it, which potentially brings a constraint to manipulations. We provide a proof for Theorem 4.3 in Appendix E.3, along with a discussion on other manipulations on word embeddings.

### 4.4 RoPE and Attention Scores: Boundaries for Q/K Manipulations

Finally, we study RoPE and the attention score computation to characterize which manipulations on Q,K matrices can preserve attention scores. After RMSNorm, inputs are fed into attention score calculations. The nonlinearity here, particularly in the softmax and RoPE functions, poses a barrier for manipulations on inputs to pass through. Hence, we follow Zeng et al. (2024) to assume that input manipulations does not change the value of attention scores. The manipulations on Q,K matrices are thus constrained under this assumption.

**Theorem 4.4** *The manipulations on Q,K matrices at layer $l$ can be categorized into*

1. *Input-related ones, passed by RMSNorm: $W_{B,i}^{(l)} = c^{-1} W_{A,i}^{(l)} PD, \quad for\ i \in \{Q,K\}$;*
2. *RoPE-related ones: $W_{B,i}^{(l)} = U_{B,i}^{(l)} W_{A,i}^{(l)}, \quad for\ i \in \{Q,K\}$;*

*where $U_{B,i}^{(l)}$ are special orthogonal matrices that keep RoPE results.*

A proof for Theorem 4.4 is provided in Appendix E.4, where we also show how manipulations on inputs are recovered after V, O matrices to satisfy Proposition 4.2. These manipulations preserve the attention score values of model A in the suspicious model B. However, they can greatly change the weights of the original model A, bringing difficulty to the development of detection methods.

### 4.5 Potential Attacks

Combining Theorem 4.3 and Theorem 4.4, the admissible manipulations on $\mathbb{W}_{A,\text{partial}}$ (word embeddings and Q, K matrices) in Definition 4.1 are restricted to

$$W_{B,i}^{(l)\top} = c^{-1} D^\top P^\top W_{A,i}^{(l)\top} U_{B,i}^{(l)\top} + E_{B,i}^{(l)\top}, \quad 1 \leq l \leq L, \ i \in \{\text{Q,K}\}, \tag{4}$$

$$W_{B,\text{emb}} = c\, W_{A,\text{emb}}\, PD + E_{B,\text{emb}}. \tag{5}$$

Here $E$ collects post-training, continual pre-training, pruning, upcycling, multimodal adaptation, or related adjustments. The nonlinear usage of $\mathbb{W}_{A,\text{partial}}$, especially through the Q, K matrices, substantially limits manipulation complexity. Consequently, checking similarity between $\mathbb{W}_{A,\text{partial}}$ and $\mathbb{W}_{B,\text{partial}}$ is typically sufficient for detection. The remaining avenues targeting $\mathbb{W}_{\mathbb{A}}/\mathbb{W}_{A,\text{partial}}$ are deferred to Appendix E.5, where we also discuss how such transformations can recover Transformer block outputs in light of Proposition 4.2.

---

[2]The conclusions generalize to models with different number of layers (possibly a result of pruning). See Appendix E.3

---

**Algorithm 1** LAP-Enhanced Unbiased Central Kernel Alignment (UCKA) Similarity Detection

---

1: Given two $L$[3]-layered LLMs $A$, $B$ and their weight matrices $\mathbb{W}_{A,\text{partial}}$, $\mathbb{W}_{B,\text{partial}}$.
2: Let $I = \text{Vocab}(A) \cap \text{Vocab}(B)$, and $m' = |I|$.
3: Let $W_{A,\text{shared-emb}} = W_{A,\text{emb}}[I,:]$ and $W_{B,\text{shared-emb}} = W_{B,\text{emb}}[I,:]$.
4: Build the cosine similarity matrix $C$ with $C_{k,l} = \dfrac{\langle (W_{A,\text{shared-emb}})_{:,k},\ (W_{B,\text{shared-emb}})_{:,l} \rangle}{\|(W_{A,\text{shared-emb}})_{:,k}\|\ \|(W_{B,\text{shared-emb}})_{:,l}\|}$.
5: Construct the permutation and signature matrices $P, D$ via LAP:
6:      Find a permutation $\pi$ maximizing $\sum_k |C_{k,\pi(k)}|$ with the Hungarian algorithm.
7:      For each column $k$, set $s_k = \text{sign}\big(C_{k,\pi(k)}\big)$.
8:      Set $P_{k,\pi(k)} = 1$ for every $k$, set $D = \text{diag}(s_1, s_2, \ldots, s_k, \ldots)$.
9: **for** $l = 1, \ldots, L$ **do**
10:      $s_Q^{(l)} = \text{UCKA}\Big(D^\top P^\top W_{A,Q}^{(l)\top}, W_{B,Q}^{(l)\top}\Big),\ s_K^{(l)} = \text{UCKA}\Big(D^\top P^\top W_{A,K}^{(l)\top}, W_{B,K}^{(l)}\Big)$
11: **end for**
12: **return** $\sum_{l=1}^{L}(|s_Q^{(l)}| + |s_K^{(l)}|)/2L$

---

## 5 METHODOLOGY

Building on Theorem 4.3 and Theorem 4.4, we propose a two-stage procedure in Algorithm 1: (i) extract $P$ and $D$ from the word embeddings; (ii) assess cross-model similarity by comparing the Q and K matrices. This design achieves fast, reliable discrimination with minimal computation.

**Extracting the Permutation and Signature from Word Embeddings**    A key property of $R_{B,\text{emb}}$ is that the permutation and signature matrices may appear in either order: both $R_{B,\text{emb}} = cPD$ and $R_{B,\text{emb}} = cDP$ are possible. However, any product $DP$ can be rewritten as $P'D'$ for some permutations $P'$ and signature matrices $D'$. Hence it suffices to recover a canonical $PD$ from the embeddings. We cast this as a Linear Assignment Problem (LAP; Burkard & Cela (1999)) solved by the Hungarian algorithm (Kuhn, 1955), which is invariant to any nonzero scalar factor. Specifically, we first restrict to the shared vocabulary and form the matrix of absolute cosine similarities between the columns of $W_{A,\text{emb}}$ and $W_{B,\text{emb}}$. Then, LAP is applied to obtain the permutation $P$. Next, we use the signs of the cosine similarities at the matched pairs to reconstruct the signature matrix $D$. This approach effectively reconstruct corresponding transformations due to its low demand of additional parameters in the detection process.

**Robust Recovery of Weight Similarities**    Despite accounting for permutation and signature manipulations, the orthogonal transformations $U_{B,i}^{(l)}$ remain challenging. They introduce a substantial nuisance parameter burden: if $W_{A,i}^{(l)} \in \mathbb{R}^{d \times n}$, then orthogonal $U_{B,i}^{(l)} \in \mathbb{R}^{d \times d}$ contributes $d^2$ parameters, which is prohibitive when $d \approx n$ and can undermine robustness by adding parameters to detections (Simmons et al., 2011). We therefore use Central Kernel Alignment (CKA) as a parameter-free similarity metric: by Theorem 3.1, CKA is invariant to orthogonal transforms and constant rescaling. While this invariance does not extend to semi-orthogonal $U_{B,i}^{(l)}$ (e.g., induced by pruning), CKA remains effective to a meaningful extent in such cases, relieving the need to explicitly reconstruct $U_{B,i}^{(l)}$. To further mitigate biases (Gretton et al., 2005; Murphy et al., 2024), we adopt the unbiased variant (Song et al., 2007), termed UCKA (see Appendix D.2).

## 6 EXPERIMENTS

In this section, we present a series of experiments designed to rigorously evaluate our proposed model fingerprinting method. We begin in Section 6.1 by verifying its fundamental ability to distinguish between derived (offspring) and independent models. Next, in Section 6.2, we assess the critical risk of false positives by comparing its performance on 90 pairs of independent models

---

[3]For LLMs with differing layer counts ($L_A$, $L_B$), e.g., from layer pruning, we find the optimal layer pairing by solving the LAP on an $L_A \times L_B$ matrix of layer-wise similarities before calculating overall similarity.

Table 1: Similarity (%) of various LLMs to LLaMA2-7B and LLaMA2-13B base models. Offspring models consistently show high similarity scores (light red), while independent models have negligible similarity (light green), demonstrating the method's discriminative power.

| Base Model: LLaMA2-7B | | | | Base Model: LLaMA2-13B | | | |
|---|---|---|---|---|---|---|---|
| Offspring | Sim | Independent | Sim | Offspring | Sim | Independent | Sim |
| WizardMath-7b | 99.99 | Baichuan-7b | 0.58 | Selfrag_llama2_13b | 99.99 | Baichuan-13b | 0.17 |
| Selfrag_7b | 99.98 | Mistral-7b | 0.63 | Nous-Hermes-13b | 99.98 | Baichuan2-13b | 0.23 |
| Vicuna-7b | 99.95 | OLMo-7b | 0.46 | Llama2-13b-orca | 99.98 | OLMo2-13b | 0.21 |
| Llama2-7b-Chat | 99.93 | Qwen-7b | 0.19 | Vicuna-13b | 99.93 | Qwen-14b | 0.20 |
| Finance-7b | 99.93 | InternLM-7b | 0.43 | Llama2-13b-Chat | 99.92 | Qwen3-14b | 0.41 |
| Llama2-7b-32K | 99.88 | MPT-7b | 0.04 | Firefly-llama2-13b | 99.81 | Jais-13b | 0.03 |
| Guanaco-7b | 99.80 | LLaMA-7b | 0.81 | Llama2-koen-13b | 97.76 | LLaMA-13b | 0.74 |
| Llama2-ko-7b | 96.79 | OpenLlama-7b | 0.71 | Llama2-13b-Estopia | 96.60 | OpenLlama-13b | 0.51 |

against two state-of-the-art baselines, HuRef and REEF. In Section 6.3, we test the method's robustness against a wide array of common post-training modifications using a comprehensive suite of 60 offspring models. Finally, in Section 6.4, we provide an overall performance comparison, leveraging ROC curves and other metrics to demonstrate our method's superior discriminative power.

**Baselines.** We compare our method against two advanced baselines: the weight-based method HuRef (Zeng et al., 2024), and the representation-based method REEF (Zhang et al., 2025).

## 6.1 Effectiveness Verification

We first established our method's core effectiveness in identifying model lineage. We collected eight offspring models for both LLaMA2-7B and LLaMA2-13B, alongside eight independent models for each size. We then calculated the similarity between the base model and these two groups.

As shown in Table 1, the results demonstrate a clear distinction. Offspring models exhibited exceptionally high similarity to their respective base models, whereas the similarity scores between independent models were negligible, forming a strong basis for intellectual property protection.

## 6.2 False Positive Risk Evaluation on 90 Unrelated Pairs

A critical requirement for any reliable fingerprinting method is an extremely low false positive rate. To evaluate this risk, we collected 10 independent 7B LLMs and 10 independent 13B LLMs, forming 45 unique pairs for each size. We then computed the pairwise similarity (%) for all 90 pairs using our method and compared the results with those from REEF and HuRef.

The heatmaps in Figure 2 illustrate our method's superior performance in avoiding false positives. For independent pairs, our method yielded mean similarity scores of just 0.49 (7B) and 0.26 (13B). These scores are nearly an order of magnitude lower than HuRef (means of 3.56 and 2.17) and two orders of magnitude lower than REEF (means of 42.47 and 47.44).

Notably, REEF frequently produces dangerously high similarity scores for unrelated models, with many pairs exceeding 80 and some even surpassing 95. Such high values could easily lead to false accusations of model theft. In contrast, the maximum similarity scores for our method were merely 1.5 (7B) and 0.8 (13B), reaffirming its significantly lower risk of false positives.

## 6.3 Robustness Verification on 60 Offspring Models

Base LLMs often undergo substantial post-training modifications, including SFT, continued pretraining, reinforcement learning (RL), pruning, upcycling, and multi-modal adaptation. These processes can significantly alter model parameters, making robustness a critical attribute for any fingerprinting technique. To rigorously evaluate our method's robustness, we curated a diverse test suite of 60 positive pairs, each consisting of a base model and a derived offspring model. This suite includes 10 pairs for each of the six modification categories listed above.

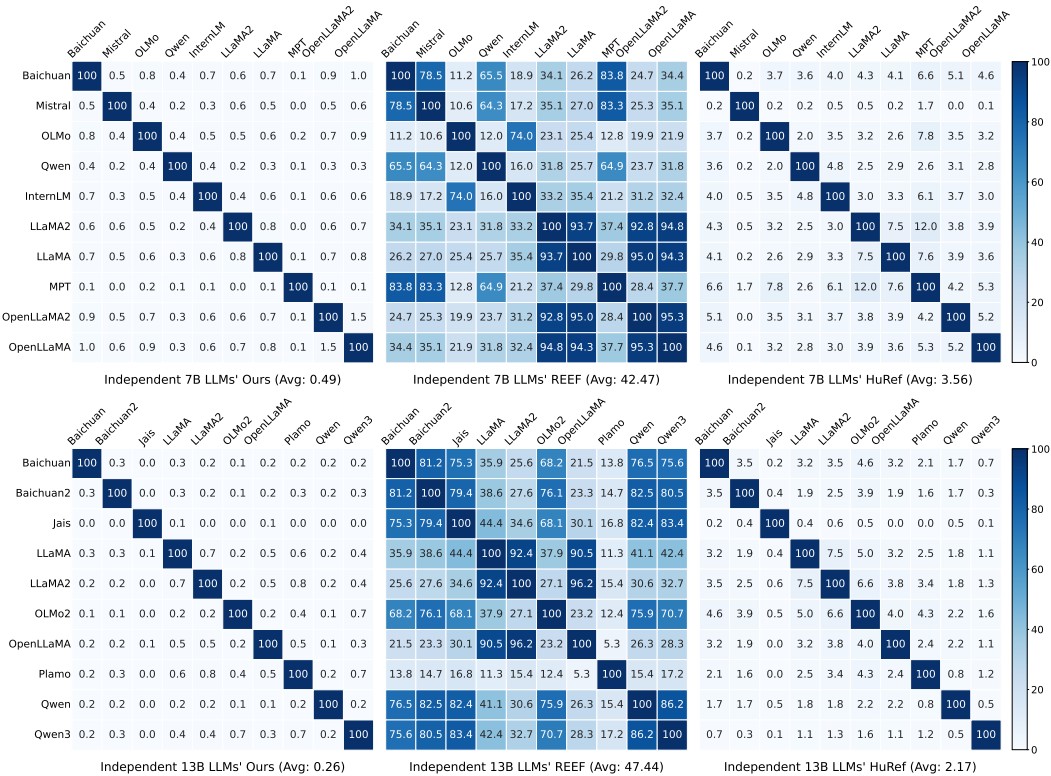

Figure 2: Pairwise similarity (%) heatmaps for independent 7B (top row) and 13B (bottom row) LLMs. The columns compare our proposed method against two baselines, REEF and HuRef. Our method consistently yields near-zero similarity scores for all independent model pairs, indicating a significantly lower risk of false positives. In contrast, REEF often produces high similarity scores ($> 80$), which could easily lead to false accusations of model theft.

The effectiveness of a fingerprinting method hinges on its ability to distinguish between related (positive) and unrelated (negative) pairs. However, absolute similarity scores (%) can be misleading. A method yielding scores of 10 for a positive pair and 0.1 for negative pairs is far more discriminative than one producing scores of 90 and 40. To accurately measure the discriminative power, we must quantify how statistically different a positive pair's similarity is from the distribution of negative pairs. For this, we use the absolute Z-score ($|Z|$)[4]. This metric measures how many standard deviations a positive pair's similarity is from the mean of the negative pairs' similarities. A larger $|Z|$ indicates that the positive pair is more likely to be a statistical outlier relative to the population of unrelated pairs and thus more highly separable.

The results, presented in Table 2, highlight the superior robustness of our method. HuRef loses effectiveness under heavy modifications like extensive continued pretraining (e.g., its $|Z|$ drops to 0.57 for Qwen2.5-coder). REEF consistently yields low $|Z|$, often below 2.0, indicating poor separability. In stark contrast, our method maintains remarkably high $|Z|$ across all 60 positive pairs, demonstrating its resilience against a wide array of demanding post-training modifications.

## 6.4 OVERALL EVALUATION

To synthesize our findings, we conducted a comprehensive evaluation against HuRef (ICS) and REEF using the full dataset of 60 positive and 90 negative model pairs. Additionally, we incorporated PCS (from HuRef) and Intrinsic Fingerprint (Yoon et al., 2025) as supplementary baselines to broaden the comparative analysis.

---

[4]In this context, the absolute Z-score is mathematically equivalent to the Mahalanobis distance.

Table 2: Absolute Z-scores (↑) of positive samples for HuRef, REEF, and our method. The full names of abbreviations and related model details are provided in [Appendix G](#).

**SFT**

| Base | Llama2-7B | | | | | | Llama2-13B | | | |
|---|---|---|---|---|---|---|---|---|---|---|
| **Offspring** | Vicuna | Finance | Selfrag | 32K | Wizard | Guanaco | Vicuna | Hermes | Estopia | Firefly |
| HuRef | 44.22 | 44.20 | 44.47 | 44.22 | 44.51 | 44.12 | 44.25 | 44.48 | 43.09 | 44.51 |
| REEF | 1.95 | 1.94 | 1.95 | 1.88 | 1.95 | 1.94 | 1.94 | 1.95 | 1.91 | 1.95 |
| Ours | 355.09 | 355.02 | 355.20 | 354.84 | 355.23 | 354.55 | 355.02 | 355.20 | 343.14 | 354.59 |

**Continual Pretrain**

| Base | Llama2-7B | | | Gemma -2B | Gemma -7B | Qwen2.5 -7B | Qwen2 -7B | | Llama2-70B | |
|---|---|---|---|---|---|---|---|---|---|---|
| **Offspring** | Llemma (700B) | Code (520B) | Python (620B) | Code (500B) | Code (500B) | Math (1T) | Coder (5.5T) | Math (700B) | Code (520B) | Python (620B) |
| HuRef | 8.15 | 9.41 | 9.19 | 28.97 | 39.51 | 0.28 | 0.57 | 1.01 | 3.43 | 2.78 |
| REEF | 1.66 | 1.31 | 1.77 | 0.32 | 1.52 | 1.04 | 1.29 | 1.09 | 1.93 | 1.92 |
| Ours | 250.32 | 250.78 | 253.28 | 198.18 | 268.72 | 177.57 | 135.17 | 183.49 | 241.12 | 232.92 |

**Upcycling**

| Base | Mistral -7B | Llama3 -8B | | | Llama2-7B | | | | | Qwen -1.8B | Minicpm -2B |
|---|---|---|---|---|---|---|---|---|---|---|---|
| **Offspring** | Mixtral | MoE v2 | MoE4 | MoE 3B | MoE2 | MoE3B -SFT | MoE2 -SFT | MoE4 -SFT | Qwen1.5 MoE | Minicpm MoE |
| HuRef | 7.97 | 42.50 | 24.58 | 21.59 | 24.91 | 21.59 | 24.91 | 24.58 | 4.03 | 11.39 |
| REEF | 1.47 | 0.47 | 0.64 | 0.64 | 0.62 | 0.63 | 0.63 | 0.62 | 0.57 | 1.49 |
| Ours | 239.01 | 332.12 | 332.23 | 326.49 | 332.12 | 326.49 | 332.12 | 332.23 | 173.36 | 150.15 |

**Multi Modal**

| Base | Llama2-7B | | Qwen2 -7B | Qwen-7B | | | Qwen2.5 -7B | Qwen2.5 -3B | Llama3 -8B | Llama2 -13B |
|---|---|---|---|---|---|---|---|---|---|---|
| **Offspring** | LLaVA | Video | VL | Audio | Audio2 | VL | VL | VL | Next | LLaVA |
| HuRef | 44.06 | 44.03 | 39.94 | 38.79 | 21.59 | 37.41 | 24.28 | 23.95 | 44.30 | 44.09 |
| REEF | 0.40 | 0.36 | 0.84 | 0.19 | 0.45 | 0.77 | 0.48 | 0.61 | 0.26 | 0.07 |
| Ours | 354.98 | 354.98 | 342.72 | 339.86 | 317.97 | 336.55 | 290.08 | 298.46 | 355.05 | 354.91 |

**RL**

| Base | Open- llama3B | Qwen2.5 -7B | Qwen2.5 -1.5B | Mixtral | Mistral-7B | | Minicpm -2B | Qwen3 -4B | Chatglm -6B | Llama3 -8B |
|---|---|---|---|---|---|---|---|---|---|---|
| **Offspring** | RLHF | Reason | Zero | DPO | DPO | Dolphin | DPO | GRPO | RLHF | DPO |
| HuRef | 44.52 | 44.58 | 44.58 | 44.57 | 44.53 | 44.54 | 44.58 | 44.58 | 44.58 | 44.52 |
| REEF | 1.94 | 1.93 | 1.94 | 1.75 | 1.48 | 1.21 | 1.92 | 1.78 | 1.96 | 1.96 |
| Ours | 355.23 | 355.27 | 355.27 | 355.23 | 355.23 | 355.23 | 355.27 | 355.27 | 355.27 | 355.23 |

**Pruning**

| Base | Llama-3-8B | | | | Llama2-7B | | | | | |
|---|---|---|---|---|---|---|---|---|---|---|
| **Offspring** | Minitron -Depth | Minitron -Width | Llama3 -1B | Llama3 -3B | Sheared 2.7B-P | Sheared 2.7B-S | Sheared 2.7B | Sheared 1.3B-P | Sheared 1.3B | Sheared 1.3B-S |
| HuRef | 28.29 | 22.23 | 0.33 | 0.73 | 22.88 | 16.00 | 15.86 | 10.06 | 7.64 | 7.64 |
| REEF | 0.52 | 0.53 | 1.15 | 0.92 | 1.75 | 1.78 | 1.77 | 1.80 | 1.79 | 1.79 |
| Ours | 344.07 | 343.00 | 12.14 | 106.29 | 328.81 | 312.44 | 312.80 | 317.79 | 297.50 | 297.50 |

As illustrated in [Figure 1](#), our method demonstrates vastly superior performance. The Receiver Operating Characteristic (ROC) curve (left) shows that our method achieves a perfect Area Under

Table 3: Detailed performance comparison of fingerprinting methods across various post-training techniques. Our method consistently achieves perfect scores (1.0) on all classification metrics (AUC, pAUC, TPR@1%FPR) and maintains a significantly larger separation margin ($\overline{|Z|}$) across all scenarios. CPT: Continual Pre-Training, UP: Upcycling, MM: Multi-modal, PR: Pruning.

| Method | Metric | SFT | CPT | UP | MM | RL | PR | All |
|---|---|---|---|---|---|---|---|---|
| **HuRef** | $\overline{|Z|} \uparrow$ | 43.748 | 10.331 | 20.805 | 36.244 | 44.559 | 13.166 | 28.142 |
| | AUC $\uparrow$ | 1.000 | 0.879 | 0.999 | 1.000 | 1.000 | 0.866 | 0.957 |
| | pAUC $\uparrow$ | 1.000 | 0.656 | 0.978 | 1.000 | 1.000 | 0.800 | 0.906 |
| | TPR@1%FPR $\uparrow$ | 1.000 | 0.500 | 0.900 | 1.000 | 1.000 | 0.800 | 0.867 |
| **REEF** | $\overline{|Z|} \uparrow$ | 1.936 | 1.384 | 0.777 | 0.443 | 1.788 | 1.381 | 1.285 |
| | AUC $\uparrow$ | 1.000 | 0.857 | 0.508 | 0.648 | 0.963 | 0.692 | 0.778 |
| | pAUC $\uparrow$ | 1.000 | 0.211 | 0.000 | 0.000 | 0.658 | 0.300 | 0.362 |
| | TPR@1%FPR $\uparrow$ | 1.000 | 0.200 | 0.000 | 0.000 | 0.600 | 0.000 | 0.300 |
| **Intrinsic Fingerprint** | $|Z| \uparrow$ | 1.535 | 1.193 | 1.408 | 1.532 | 1.542 | 1.141 | 1.392 |
| | AUC $\uparrow$ | 1.000 | 0.896 | 0.969 | 1.000 | 1.000 | 0.876 | 0.957 |
| | pAUC $\uparrow$ | 1.000 | 0.422 | 0.800 | 1.000 | 1.000 | 0.400 | 0.770 |
| | TPR@1%FPR $\uparrow$ | 1.000 | 0.300 | 0.800 | 1.000 | 1.000 | 0.400 | 0.750 |
| **PCS** | $|Z| \uparrow$ | 73.786 | 74.650 | 0.727 | 14.950 | 163.399 | 17.226 | 57.457 |
| | AUC $\uparrow$ | 0.958 | 0.959 | 0.791 | 0.802 | 0.984 | 0.666 | 0.860 |
| | pAUC $\uparrow$ | 0.656 | 0.600 | 0.078 | 0.500 | 0.900 | 0.100 | 0.472 |
| | TPR@1%FPR $\uparrow$ | 0.500 | 0.600 | 0.000 | 0.500 | 0.900 | 0.100 | 0.433 |
| **Ours** | $\overline{|Z|} \uparrow$ | 353.788 | 219.155 | 287.634 | 334.556 | 355.250 | 267.233 | 302.936 |
| | AUC $\uparrow$ | 1.000 | 1.000 | 1.000 | 1.000 | 1.000 | 1.000 | 1.000 |
| | pAUC $\uparrow$ | 1.000 | 1.000 | 1.000 | 1.000 | 1.000 | 1.000 | 1.000 |
| | TPR@1%FPR $\uparrow$ | 1.000 | 1.000 | 1.000 | 1.000 | 1.000 | 1.000 | 1.000 |

the Curve (AUC) of 1.0, indicating flawless discrimination. This starkly outperforms both HuRef ($AUC = 0.957$) and REEF ($AUC = 0.778$).

In practical applications, preventing false accusations of model theft is paramount, making performance at a very low False Positive Rate (FPR) essential. We therefore employ two stricter metrics: the partial AUC for FPR $< 0.05$ (pAUC) and the True Positive Rate at a 1% FPR (TPR@1%FPR).

Table 3 details the performance breakdown across post-training techniques. Our method consistently achieves perfect scores (1.0) on all classification metrics (AUC, pAUC, and TPR@1%FPR) across all categories. In contrast, the baseline methods show significant limitations. REEF's performance collapses in several scenarios, with pAUC and TPR@1%FPR scores falling to 0.0 for Upcycling and Multi-modal. HuRef also shows vulnerability, with its TPR@1%FPR dropping to 0.500 under Continual Pre-Training. Our method's perfect classification scores, combined with its substantially larger average absolute Z-score ($\overline{|Z|}$), underscore its superior robustness and reliability.

# 7 CONCLUSIONS

In this paper, we propose a training-free fingerprinting method for LLM identification. Our approach does not impair LLM's general capability while exhibiting robustness against fine-tuning, extensive continued pretraining, reinforcement learning, multimodal extension, pruning, and upcycling, and simultaneously avoids the risk of false positives. Experiments on a testbed comprising 150 pairs of LLMs demonstrate the effectiveness of our method.

# 8 ACKNOWLEDGEMENTS

This work is sponsored by the National Key Research and Development Program of China (No. 2023ZD0121402) and the National Natural Science Foundation of China (NSFC) grant (No. 62576211).

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

## A    ETHICS STATEMENT

The primary motivation for this work is to protect the intellectual property of large language models (LLMs), thereby promoting fairness and accountability within the AI community. We acknowledge the potential societal impact of such a technology, particularly the risk of false accusations of model theft. Therefore, a core design principle of our method is to achieve an exceptionally low false-positive rate, as empirically demonstrated in our experiments. By establishing a high-fidelity verification system, we aim to foster a more transparent and trustworthy open-source ecosystem. All models used in this study are publicly available, and our research does not involve any private or sensitive user data.

## B    REPRODUCIBILITY STATEMENT

To ensure the reproducibility of our results, we have included the complete source code in the supplementary material. We also provided a detailed description of our methodology in Section 5, including a step-by-step algorithm. All models used for evaluation are publicly available from sources such as the Hugging Face Hub, and a comprehensive list mapping our model abbreviations to their full names is included in Appendix G. A public code repository will be made available upon publication.

## C    THE USE OF LARGE LANGUAGE MODELS

The core research and analysis presented in this manuscript were conducted without the use of Large Language Models (LLMs). An LLM was utilized exclusively to improve the language and clarity of the text.

# D   DEFINITIONS AND NOTATIONS

## D.1   LLM ARCHITECTURE

**Weights**   Here we provide notations for the rest of model A's weights. In addition to the notations in Section 3, we further denote s, $W_{A,\text{lm}}$ as the language model head, and $W_{A,\text{V}}^{(l)}, W_{A,\text{O}}^{(l)}, \mathbb{W}_{A,\text{ffn}}^{(l)}$ the V,O matrices and FFN at the $l$-th layer, and

$$\mathbb{W}_{A,\text{others}} = \mathbb{W}_A / \mathbb{W}_{A,\text{partial}}$$

$$= \{W_{A,\text{lm}}\} \bigcup_{l=1}^{L} \mathbb{W}_{A,\text{ffn}}^{(l)} \bigcup_{l=1}^{L} \{W_{A,i}^{(l)} \mid i \in \{\text{V}, \text{O}\}\}$$

as the rest of model A's weights.

**Functions**   Here we provide definitions for functions in an LLM.

**Definition D.1 (RMSNorm)**   *Given the input matrix $X \in \mathbb{R}^{m \times n}$ of model A, the learnable parameters $\omega_A^{(l)} = (\omega_{A,1}^{(l)}, \dots, \omega_{A,n}^{(l)}) \in \mathbb{R}^n$ at the $l$-th layer, and a constant $\epsilon_A \in \mathbb{R}$ (this constant is usually shared across all layers of an LLM), RMSNorm at layer $l$ is a function that*

$$\text{RMSNorm}(X, \omega_A^{(l)}, \epsilon_A) = \text{diag}^{-\frac{1}{2}}\left((X \odot X)\mathbf{1}_n + \epsilon_A \mathbf{1}_m\right) X \ \text{diag}(\omega_A^{(l)}). \tag{6}$$

**Definition D.2 (Self-Attention with RoPE)**   *Given an input $X$ to model A, the self-attention mechanism at layer $l$ is a function $f_{attn}$ that*

$$f_{attn}^{(l)}(X) = \text{AttnScore}(X, W_{A,Q}^{(l)}, W_{A,K}^{(l)}, \theta) \cdot X W_{A,V}^{(l)\top} \cdot W_{A,O}^{(l)\top} \tag{7}$$

*where AttnScore is the attention score function with RoPE, i.e.*

$$\text{AttnScore}(X, W_{A,Q}^{(l)}, W_{A,K}^{(l)}, \theta) = \text{softmax}\left(\frac{1}{\sqrt{d}} RoPE(X W_{A,Q}^{(l)\top}, \theta) \cdot RoPE(X W_{A,K}^{(l)\top}, \theta)^\top\right), \tag{8a}$$

$$RoPE(X W_{A,Q}^{(l)\top}, \theta) = [x_1^\top W_{A,Q}^{(l)\top} R_\theta(0), \dots, x_m^\top W_{A,Q}^{(l)\top} R_\theta(m-1)], \tag{8b}$$

$$RoPE(X W_{A,K}^{(l)\top}, \theta) = [x_1^\top W_{A,K}^{(l)\top} R_\theta(0), \dots, x_m^\top W_{A,K}^{(l)\top} R_\theta(m-1)]. \tag{8c}$$

*Here $x_i^\top \in \mathbb{R}^n$ is the $i$-th row of $X$, $\theta$ is the frequency base for RoPE and $R_\theta$ is the rotation matrix (Su et al., 2024).*

RoPE's definition in Definition D.2 is slightly different from their implementations, but still an equivalent version to Su et al. (2024).

**Definition D.3 (Feed-forward Networks)**   *Given the input $X$, the feed-forward network which adopts the SwiGLU MLP is the function $f_{ffn}^{(l)}$ defined by*

$$f_{ffn}^{(l)}(X) = \left(SiLU\left(X W_{A,gate}^{(l)\top}\right) \odot \left(X W_{A,up}^{(l)\top}\right)\right) W_{A,down}^{(l)\top},$$

*where $W_{A,up}^{(l)}, W_{A,gate}^{(l)}, W_{A,down}^{(l)}$ are bias-free weight matrices, $\odot$ denotes the Hadamard product, and $SiLU(z) = z\,\sigma(z)$ with $\sigma(z) = \frac{1}{1+e^{-z}}$. We write $\mathbb{W}_{A,ffn}^{(l)} = \{W_{A,up}^{(l)}, W_{A,gate}^{(l)}, W_{A,down}^{(l)}\}$.*

We abuse the notations in Definition D.1 by denoting $\text{RMSNorm}_{\text{attn}}^{(l)}(X) \triangleq \text{RMSNorm}(X, \omega_{A,\text{attn}}^{(l)}, \epsilon_A)$ as the attention norm and $\text{RMSNorm}_{\text{ffn}}^{(l)}(X) \triangleq \text{RMSNorm}(X, \omega_{A,\text{ffn}}^{(l)}, \epsilon_A)$ the ffn norm at layer $l$ of model A. Then, the definition of a Transformer block at layer $l$ of model A can be summartized as follows.

**Definition D.4 (Transformer Block)**   *Combining all the components with residual connections, the transformer block at the $l$-th layer of model A is a function $f_{Transformer}^{(l)}$ that*

$$f_{Transformer}^{(l)} = \left(I + f_{ffn}^{(l)} \circ RMSNorm_{ffn}^{(l)}\right) \circ \left(I + f_{attn}^{(l)} \circ RMSNorm_{attn}^{(l)}\right). \tag{9}$$

where $I$ is the identity mapping, $f_{attn}^{(l)}$ and $f_{ffn}^{(l)}$ are the attention and FFN at $l$-th layer respectively, $RMSNorm_{ffn}^{(l)}$ and $RMSNorm_{attn}^{(l)}$ are RMSNorm functions for these two components.

**Definition D.5 (LLM Architecture)** *Given an word embedding $X$, the last hidden state $Y$ of an $L$-layered LLM A is*

$$Y = f_{Transformer}^{(L)} \circ f_{Transformer}^{(L-1)} \circ \cdots \circ f_{Transformer}^{(L-1)}(X) \cdot W_{A,lm}^\top. \tag{10}$$

## D.2 CENTRAL KERNEL ALIGNMENT (CKA)

Given two input matrices $X_1 \in \mathbb{R}^{m \times n_1}$ and $X_2 \in \mathbb{R}^{m \times n_2}$, CKA is a function mapping paired input matrices to $[0, 1]$, and

$$\text{CKA}(X_1, X_2) = \frac{\text{HSIC}(K_{X_1}, K_{X_2})}{\sqrt{\text{HSIC}(K_{X_1}, K_{X_1}) \cdot \text{HSIC}(K_{X_2}, K_{X_2})}} \tag{11}$$

where $\text{HSIC}(K_{X_1}, K_{X_2}) = \frac{1}{(m-1)^2} \text{tr}(K_{X_1} H_m K_{X_2} H_m)$, $(K_{X_1})_{ij} = k((X_1)_i, (X_1)_j)$, $(K_{X_2})_{ij} = k((X_2)_i, (X_2)_j)$ are kernel matrices with the kernel function $k(\cdot, \cdot) : \mathbb{R}^m \times \mathbb{R}^m \to \mathbb{R}_+$, and $H_m = I_m - \frac{1}{m} 1_m 1_m^\top$ is the centering matrix. The linear kernels give $K_X = XX^\top$ for a matrix $X$. To reduce the finite-sample bias of Eq. (6), we use the unbiased HSIC estimator (Song et al., 2007). Let $\tilde{K}_{X_i}$ be $K_{X_i}$ with its diagonal set to zero, i.e., $(\tilde{K}_{X_i})_{ii} = 0$. With $1_m$ the all-ones vector of length $m$, the unbiased HSIC is

$$\text{HSIC}_u(\tilde{K}_{X_1}, \tilde{K}_{X_2}) = \frac{1}{m(m-3)} \left[ \text{tr}(\tilde{K}_{X_1} \tilde{K}_{X_2}) + \frac{(1_m^\top \tilde{K}_{X_1} 1_m)(1_m^\top \tilde{K}_{X_2} 1_m)}{(m-1)(m-2)} - \frac{2}{m-2} 1_m^\top \tilde{K}_{X_1} \tilde{K}_{X_2} 1_m \right]. \tag{12}$$

We then define

$$\text{UCKA}(X_1, X_2) = \frac{\text{HSIC}_u(\tilde{K}_{X_1}, \tilde{K}_{X_2})}{\sqrt{\text{HSIC}_u(\tilde{K}_{X_1}, \tilde{K}_{X_1}) \cdot \text{HSIC}_u(\tilde{K}_{X_2}, \tilde{K}_{X_2})}}, \tag{13}$$

which preserves Theorem 3.1 and yields values in $[-1, 1]$.

# E PROOFS AND DISCUSSIONS

## E.1 PROOF FOR THEOREM 3.1

We give the proof based on the definition of CKA with linear kernels in Appendix D.2. Given input matrices $X_1, X_2$, orthogonal matrices $U_1, U_2$ and constants $c_1, c_2$, the kernel functions yield

$$K_{c_1 X_1 U_1} = c_1 X_1 U_1 U_1^\top X_1^\top c_1 = c_1^2 X_1 X_1^\top = c_1^2 K_{X_1},$$
$$K_{c_2 X_2 U_2} = c_2 X_2 U_2 U_2^\top X_2^\top c_2 = c_2^2 X_2 X_2^\top = c_2^2 K_{X_2}.$$

Therefore, corresponding HSIC results are

$$\text{HSIC}(K_{c_1 X_1 U_1}, K_{c_2 X_2 U_2}) = \frac{c_1^2 c_2^2}{(m-1)^2} \text{tr}(K_{X_1} H_m K_{X_2} H_m) = c_1^2 c_2^2 \text{HSIC}(K_{X_1}, K_{X_2}),$$

$$\text{HSIC}(K_{c_1 X_1 U_1}, K_{c_1 X_1 U_1}) = \frac{c_1^4}{(m-1)^2} \text{tr}(K_{X_1} H_m K_{X_1} H_m) = c_1^4 \text{HSIC}(K_{X_1}, K_{X_1}),$$

$$\text{HSIC}(K_{c_2 X_2 U_2}, K_{c_2 X_2 U_2}) = \frac{c_2^4}{(m-1)^2} \text{tr}(K_{X_2} H_m K_{X_2} H_m) = c_2^4 \text{HSIC}(K_{X_2}, K_{X_2}).$$

Hence, it follows that

$$\begin{aligned}
\text{CKA}(c_1 X_1 U_1, c_2 X_2 U_2) &= \frac{\text{HSIC}(K_{c_1 X_1 U_1}, K_{c_2 X_2 U_2})}{\sqrt{\text{HSIC}(K_{c_1 X_1 U_1}, K_{c_1 X_1 U_1}) \cdot \text{HSIC}(K_{c_2 X_2 U_2}, K_{c_2 X_2 U_2})}} \\
&= \frac{c_1^2 c_2^2 \text{HSIC}(K_{X_1}, K_{X_2})}{c_1^2 c_2^2 \sqrt{\text{HSIC}(K_{X_1}, K_{X_1}) \cdot \text{HSIC}(K_{X_2}, K_{X_2})}} \\
&= \frac{\text{HSIC}(K_{X_1}, K_{X_2})}{\sqrt{\text{HSIC}(K_{X_1}, K_{X_1}) \cdot \text{HSIC}(K_{X_2}, K_{X_2})}} \\
&= \text{CKA}(X_1, X_2).
\end{aligned}$$

## E.2 PROOF FOR PROPOSITION 4.2

The goal is to show the manipulation $T$ commutes with $f$, i.e. $f \circ T = T \circ f$. If $f = f_n \circ \cdots \circ f_1$, then it suffices to show

$$f_n \circ \cdots \circ f_1 \circ T = T \circ f_n \circ \ldots f_1.$$

Since

$$f_1 \circ T = T \circ f_1, f_2 \circ T = T \circ f_2, \ldots, f_n \circ T = T \circ f_n,$$

It is direct that

$$f \circ T = T \circ f.$$

## E.3 PROOF FOR THEOREM 4.3 AND DISCUSSIONS ON OTHER INPUT MANIPULATIONS

**We first show a proof for the claims in Theorem 4.3.** The core lies in preserving the diagonal structure of RMSNorm parameters in Definition D.1. Given the input $X$ to layer $l$, the manipulations change it to $cXPD$ and fed the manipulated input into the attention norm of model B. In order to recover the manipulations on inputs after RMSNorm, i.e. for any manipulation $T$ in the input $X$ (an example is $T(X) = cXPD$),

$$\text{RMSNorm}_{\text{attn}}^{(l)} \circ T = T \circ \text{RMSNorm}_{\text{attn}}^{(l)}, \tag{14}$$

it suffices to show

$$\exists \omega_{B,\text{attn}}^{(l)} \text{ and } \epsilon_B, \text{ s.t. RMSNorm}(X, \omega_{A,\text{attn}}^{(l)}, \epsilon_A) \cdot cPD = \text{RMSNorm}(cXPD, \omega_{B,\text{attn}}^{(l)}, \epsilon_B)$$

since the learnable parameters and norm epsilon can be modified to satisfy Equation 14 and Proposition 4.2. By setting $\epsilon_B = c^2 \epsilon_A$ and $\text{diag}(\omega_{B,\text{attn}}^{(l)}) = cD^\top P^\top \text{diag}(\omega_{A,\text{attn}}^{(l)})PD$, one has

$$
\begin{aligned}
&\text{RMSNorm}(cXPD, \omega_{B,\text{attn}}^{(l)}, \epsilon_B) \\
&= \text{diag}^{-\frac{1}{2}}\left(\left((cXPD) \odot (cXPD)\right)\mathbf{1}_n + \epsilon_B \mathbf{1}_m\right) \cdot cXPD \cdot \text{diag}(\omega_{B,\text{attn}}^{(l)}) \\
&= \text{diag}^{-\frac{1}{2}}\left(c^2(X \odot X)\mathbf{1}_n + c^2 \epsilon_A \mathbf{1}_m\right) \cdot cXPD \cdot cD^\top P^\top \text{diag}(\omega_{A,\text{attn}}^{(l)})PD \\
&= c \cdot \text{diag}^{-\frac{1}{2}}\left((X \odot X)\mathbf{1}_n + \epsilon_A \mathbf{1}_m\right) \cdot X \cdot \text{diag}(\omega_{A,\text{attn}}^{(l)}) \cdot PD \\
&= \text{RMSNorm}(X, \omega_{A,\text{attn}}^{(l)}) \cdot cPD.
\end{aligned}
$$

The proof still holds for partial permutations $P$, i.e. $P$ is rectangular with number of rows more than number of columns, since $cD^\top P^\top \text{diag}(\omega_{A,\text{attn}}^{(l)})PD$ is still a diagonal matrix. Therefore, $R_{B,\text{emb}} = cPD$ under Proposition 4.2. **However, for more general manipulations, Proposition 4.2 does not hold for RMSNorm.** We give two examples for illustration.

First, **orthogonal transformations on the input generally fail to pass through RMSNorm**. Given any orthogonal matrices $U$ as the manipulation on $X$, i.e., $T(X) = XU$, if one attempts to keep Equation 14, then it requires

$$\exists \omega^{(l)} \text{ and } \epsilon, \text{ s.t. } \text{RMSNorm}(XU, \omega^{(l)}, \epsilon) = \text{RMSNorm}(X, \omega_{A,\text{attn}}^{(l)}, \epsilon_A)U.$$

However, since $(X \odot X)\mathbf{1}_n = (XU \odot XU)\mathbf{1}_n$, we have

$$\text{RMSNorm}(XU, \omega, \epsilon) = \text{diag}^{-\frac{1}{2}}\left((X \odot X)\mathbf{1}_n + \epsilon \mathbf{1}_m\right) \cdot XU \cdot \text{diag}(\omega^{(l)}).$$

Hence, it is required that

$$U \cdot \text{diag}(\omega^{(l)}) = \text{diag}(\omega_{A,\text{attn}}^{(l)})U$$

which suggests $U^\top \text{diag}(\omega_{A,\text{attn}}^{(l)})U$ is a diagonal matrix. Nevertheless, this property generally does not hold if $U$ is not a (partial) permutation matrix $P$ or a signature matrix $D$.

**Second, non-orthogonal transformations on inputs generally fail to pass through RMSNorm.** Given an arbitrary (invertible) transformation $M$, Proposition 4.2 and Equation 14 require that

$$\exists \omega^{(l)} \text{ and } \epsilon, \text{ s.t. } \text{RMSNorm}(XM, \omega^{(l)}, \epsilon) = \text{RMSNorm}(X, \omega_{A,\text{attn}}^{(l)}, \epsilon_A)M.$$

However, since

$$\text{RMSNorm}(XM, \omega^{(l)}, \epsilon) = \text{diag}^{-\frac{1}{2}}\left((XM \odot XM)\mathbf{1}_n + \epsilon \mathbf{1}_m\right) \cdot XM \cdot \text{diag}(\omega^{(l)}),$$

any $M$ that do not satisfy $MM^\top = c'I$ where $c'$ is a constant and $I$ is the identity matrix can hardly recover the manipulations due to the nonlinearity in norm functions. Moreover, similar to the case with orthogonal transformations, $M \cdot \text{diag}(\omega^{(l)}) = \text{diag}(\omega_{A,\text{attn}}^{(l)})M$ also requires $M$ to be (partial) permutation or signature matrices (or a combination of the both).

**Third, we additionally clarify that although the combination of (partial) permutation and signature matrices can be in any order, it is reasonable to fix it as $R_{B,\text{emb}} = cPD$.** To be specific, we clarify that for any (partial) permutation matrix $P'$ and signature matrix $D'$, there exists a (partial) permutation matrix $P$ and a signature $D$ such that

$$D'P' = PD$$

The proof is straightforward. Define $P$ entry-wise by

$$P_{ij} := \left|(D'P')_{ij}\right| \in \{0, 1\}.$$

Since left-multiplication by $D'$ only flips signs, each column of $D'P'$ has at most one nonzero entry. Hence, $P$ is a partial permutation matrix. For each column $j$, if that column of $D'P'$ has its unique nonzero at row $i$, set

$$D_{jj} := \text{sgn}\left((D'P')_{ij}\right) \in \{\pm 1\}.$$

If the column is entirely zero, choose $D_{jj} \in \{\pm 1\}$ arbitrarily. Then for all $i, j$,

$$(PD)_{ij} = P_{ij} D_{jj} = \left| (D'P')_{ij} \right| \cdot \text{sgn}\big( (D'P')_{ij} \big) = (D'P')_{ij},$$

where we interpret $\text{sgn}(0) = 1$. Therefore, $PD = D'P'$.

**Last, we clarify that the conclusions in Theorem 4.3 can generalize to the case where model A and model B may not share the same architecture.** It is a direct result from the fact that the two LLMs have shared tokens and that the pruning over dimensions of a model's word embeddings can be viewed as a partial permutation matrix.

### E.4    PROOF FOR THEOREM 4.4 AND RELATED DISCUSSIONS

**We first provide the proof for Theorem 4.4 based on Definition D.2.**

We denote by $X$ the original input to self-attention, and $X'$ the manipulated one. Then, Theorem 4.3 suggests that

$$X' = cXPD$$

Therefore, the attention score of model B becomes

$$\text{AttnScore}(cXPD, W_{B,Q}^{(l)}, W_{B,K}^{(l)}, \theta) = \text{softmax}\Big( \tfrac{1}{\sqrt{d}} \text{RoPE}(cXPDW_{B,Q}^{(l)\top}, \theta) \cdot \text{RoPE}(cXPDW_{B,K}^{(l)\top}, \theta)^\top \Big).$$

To keep attention scores of model B identical to that of model A, it is required that

$$\text{RoPE}(cXPDW_{B,Q}^{(l)\top}, \theta) \cdot \text{RoPE}(cXPDW_{B,K}^{(l)\top}, \theta)^\top = \text{RoPE}(XW_{A,Q}^{(l)\top}, \theta) \cdot \text{RoPE}(XW_{A,K}^{(l)\top})^\top.$$

By Equation 8b and Equation 8c, this translates into

$$c^2 x_i^\top PDW_{B,Q}^{(l)\top} R_\theta(i) R_\theta^\top(j) W_{B,K}^{(l)} D^\top P^\top x_j = x_i^\top W_{A,Q}^{(l)\top} R_\theta(i) R_\theta^\top(j) W_{A,K}^{(l)} x_j$$

Therefore, the recovery of model A's attention score requires column-wise transformations to eliminate the **input-related manipulations passed by RMSNorm**, i.e.

$$W_{B,Q}^{(l)} = c^{-1} W_{A,Q}^{(l)} PD, \quad W_{B,K}^{(l)} = c^{-1} W_{A,K}^{(l)} PD \tag{15}$$

which suggests

$$R_{B,Q}^{(l)} = R_{B,K}^{(l)} = c^{-1} PD.$$

Although Equation 15 recovers attention scores, the Q,K matrices can still be modified without changing attention scores. An example is, given a rotation matrix $R = R_\theta(k)$ with the same frequency base as RoPE, multiplying Q,K matrices with it does not change the attention scores, i.e.

$$x_i^\top W_{B,Q}^{(l)\top} R_\theta(k) R_\theta(i) R_\theta^\top(j) R_\theta^\top(k) W_{B,K}^{(l)} x_j = x_i^\top W_{B,Q}^{(l)\top} R_\theta(i+k) R_\theta^\top(j+k) W_{B,K}^{(l)\top} x_j$$
$$= x_i^\top W_{B,Q}^{(l)\top} R_\theta(i-j) W_{B,K}^{(l)} x_j$$
$$= x_i^\top W_{B,Q}^{(l)\top} R_\theta(i) R_\theta^\top(j) W_{B,K}^{(l)\top} x_j.$$

Furthermore, this conclusion can be generalize to any rotation matrix with structures similar to RoPE rotation matrices. Given $R = \text{diag}(R(\psi_0), R(\psi_1), \dots)$ with $R(\psi_k) = \begin{bmatrix} \cos \psi_k & -\sin \psi_k \\ \sin \psi_k & \cos \psi_k \end{bmatrix}$, multiplying Q,K matrices by $R$ keeps the attention scores since

$$RR_\theta(i) R_\theta^\top(j) R^\top = R_\theta(i) R_\theta^\top(j). \tag{16}$$

Hence, there are various rotation matrices to significantly change Q,K matrices but preserve attention scores. Since these rotation matrices are naturally orthogonal, we reformulate the row-wise transformations on Q,K matrices $L_{B,Q}, L_{B,K}$ as $U_{B,Q}^{(l)}$ and $U_{B,K}^{(l)}$.

Therefore, we denote **RoPE-related manipulations as**

$$W_{B,Q}^{(l)} = U_{A,Q}^{(l)} W_{B,Q}^{(l)}, \quad W_{B,K}^{(l)} = U_{A,K}^{(l)} W_{B,K}^{(l)}.$$

On the other hand, semi-orthogonal $U_{B,\mathrm{Q}}^{(l)}, U_{B,\mathrm{K}}^{(l)}$ are also possible. This is because of the common practice in pruning may select different rows of Q,K matrices, which suggests multiplying Q,K matrices with row-wise partial permutations.

**Next, we show how Proposition 4.2 is satisfied at the self-attention mechanism based on Theorem 4.4.** Let $W_{B,\mathrm{V}}^{(l)} = W_{A,\mathrm{V}}^{(l)} PD$ and $W_{B,\mathrm{O}}^{(l)} = D^\top P^\top W_{A,\mathrm{O}}^{(l)}$. Then,

$$
\begin{aligned}
X' W_{B,\mathrm{V}}^{(l)\top} W_{B,\mathrm{O}}^{(l)\top} &= cXPD \cdot D^\top P^\top W_{A,\mathrm{V}}^{(l)\top} W_{A,\mathrm{O}}^{(l)\top} PD \\
&= X W_{A,\mathrm{V}}^{(l)\top} W_{A,\mathrm{O}}^{(l)\top} \cdot cPD
\end{aligned}
$$

recovers the manipulation over inputs in Equation 7.

### E.5 HOW POTENTIAL ATTACKS IN SECTION 4.5 RECOVERS THE OUTPUT OF TRANSFORMERS

We provide an illustration based on Definition D.3, Definition D.4 and Definition D.5. Previous parts have demonstrated how the manipulations on inputs pass through self-attention. Hence, it suffices to show how the manipulations pass through FFN and yield an output identical to model A after the language model head.

For layer $l$, let $W_{B,\mathrm{gate}}^{(l)} = c^{-1} W_{A,\mathrm{gate}}^{(l)} PD, W_{B,\mathrm{gate}}^{(l)} = c^{-1} W_{A,\mathrm{gate}}^{(l)} PD, W_{B,\mathrm{down}}^{(l)} = cD^\top P^\top W_{A,\mathrm{down}}^{(l)}$. We denote $X' = cXPD$ as the manipulated input. Then for model B,

$$
\begin{aligned}
& f_{\mathrm{ffn}}^{(l)}(X') \\
&= \Big( \mathrm{SiLU}\big(X' W_{B,\mathrm{gate}}^{(l)\top}\big) \odot \big(X' W_{B,\mathrm{up}}^{(l)\top}\big) \Big) W_{B,\mathrm{down}}^{(l)\top} \\
&= \Big( \mathrm{SiLU}\big(cXPD \cdot D^\top P^\top W_{A,\mathrm{gate}}^{(l)\top} \cdot c^{-1}\big) \odot \big(cXPD \cdot D^\top P^\top W_{A,\mathrm{up}}^{(l)\top} \cdot c^{-1}\big) \Big) W_{B,\mathrm{down}}^{(l)\top} \cdot cPD \\
&= \Big( \mathrm{SiLU}\big(X W_{A,\mathrm{gate}}^{(l)\top}\big) \odot \big(X W_{A,\mathrm{up}}^{(l)\top}\big) \Big) W_{A,\mathrm{down}}^{(l)\top} \cdot cPD.
\end{aligned}
$$

By Definition D.4 and Definition D.5, the manipulation is propagated through transformer layers. At the language model head, we denote $W_{B,\mathrm{lm}} = c^{-1} W_{A,\mathrm{lm}} PD$ and abuse $X' = cXPD$ as the manipulated input. Then,

$$
\begin{aligned}
X' W_{B,\mathrm{lm}}^{(l)\top} &= cXPD \cdot D^\top P^\top W_{A,\mathrm{lm}}^{(l)\top} \cdot c^{-1} \\
&= X W_{A,\mathrm{lm}}^{(l)\top}
\end{aligned}
$$

recovers the output of model A.

## F    ABLATION STUDIES

### F.1    NUMBER OF OVERLAPPING VOCABULARY TOKENS

Although Algorithm 1 uses overlapping tokens to recover the signature matrices and permutations, the detection performance of AWM does not heavily depend on the amount of vocabulary overlap. In fact, AWM remains effective even when only a small number of tokens ( 100 tokens) are shared between the two vocabularies. To quantify this, we conduct an ablation study on the number of overlapping tokens used in Algorithm 1. Specifically, for each scenario in Table 2 (SFT, Continual Pretraining, Upcycling, Multi-Modal, RL, and Pruning), we compute the average absolute Z-score under different numbers of overlapping vocabulary tokens and report the results in Figure 3.

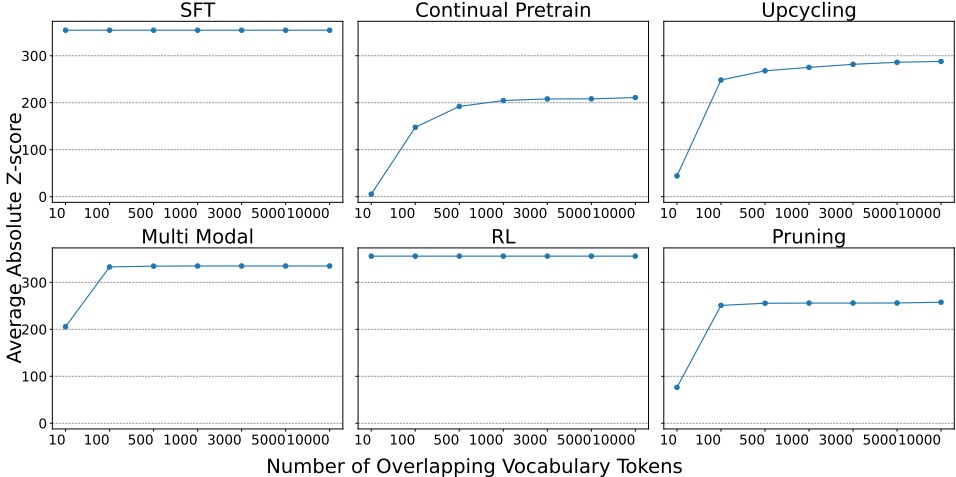

Figure 3: Ablation studies on the number of overlapping vocabulary tokens vs AWM's average absolute Z-score in Table 2. AWM remain effective even when there are only 100 overlapping tokens used in Algorithm 1.

### F.2    CKA ABLATIONS

We further conduct an ablation study on the design of CKA in Algorithm 1. In AWM, we use the unbiased version with linear kernel for computation efficiency and accuracy. Here we investigate into more variants of CKA, and summarize the results in Table 4. It can be seen that the choice of unbiasedness is crucial to the robustness of our method. Meanwhile, although RBF kernels show stronger performance in Table 4, the choice of linear kernel has already yield strong performance, which we choose for better computational efficiency.

| CKA Kernel | Linear | | RBF | |
|---|---|---|---|---|
| CKA Biasedness | Unbiased | Biased | Unbiased | Biased |
| SFT | 356.0223 | 18.3099 | 527.3949 | 11.5843 |
| Continual Pretrain | 217.5003 | 11.2954 | 320.9173 | 7.0701 |
| Upcycling | 291.6191 | 15.1994 | 432.1476 | 9.6776 |
| Multi Modal | 336.6757 | 17.3766 | 498.8918 | 11.0056 |
| RL | 357.5001 | 18.3850 | 529.6448 | 11.6330 |
| Pruning | 268.9175 | 13.8985 | 394.8391 | 8.7331 |

Table 4: AWM's average absolute Z-scores under CKA with different kernels and biasedness.

# G   MODEL DETAILS

Table 5 details the specifications of the offspring models analyzed in our study, mapping abbreviations to their base models and training datasets. To facilitate reproducibility, each entry in the "Full Model Name" column serves as a direct link to the official model checkpoint hosted on the Hugging Face Hub.

Table 5: Mapping of model abbreviations to their full model names, corresponding Hugging Face checkpoints url, base models, and relevant training corpus information.

| Abbreviation | Base Model | Full Model Name | Train Corpus |
|---|---|---|---|
| Vicuna | Llama2-7B | vicuna-7b-v1.5 | ShareGPT |
| Selfrag | Llama2-7B | selfrag_llama2_7b | Self-RAG Data |
| 32K | Llama2-7B | LLaMA-2-7B-32K | Book, ArXiv, etc. |
| Wizard | Llama2-7B | WizardMath-7B-V1.0 | GSM8k |
| Guanaco | Llama2-13B | llama-2-7b-guanaco | OpenAssistant |
| Vicuna | Llama2-13B | vicuna-13b-v1.5 | ShareGPT |
| Hermes | Llama2-13B | Nous-Hermes-Llama2-13b | GPTeacher, WizardLM, etc. |
| Estopia | Llama2-13B | LLaMA2-13B-Estopia | EstopiaV9/V13, Tiefighter, etc. |
| Finance | Llama2-7B | llama-2-7b-finance | Financial Dataset |
| Firefly | Llama2-13B | firefly-llama2-13b | CLUE, ThucNews, etc. |
| Llemma | Llama2-7B | llemma_7b | ArXiv, OpenWebMath, etc. |
| Code | Llama2-7B | CodeLlama-7b-hf | Deduped code, Natural language |
| Python | Llama2-7B | CodeLlama-7b-Python-hf | Python code |
| Code | Gemma-2B | codegemma-2b | Math, Synthetic code, etc. |
| Code | Gemma-7B | codegemma-7b | Code, Natural language |
| Math | Qwen2.5-7B | Qwen2.5-Math-7B | Web, Books, etc. |
| Coder | Qwen2.5-7B | Qwen2.5-Coder-7B | Source Code, Synthetic data, etc. |
| Math | Qwen2-7B | Qwen2-Math-7B | Math Data |
| Code | Llama2-70B | CodeLlama-70b-hf | Deduped code, Natural language |
| Python | Llama2-70B | CodeLlama-70b-Python-hf | Python code |
| Mixtral | Mistral-7B | Nous-Hermes-2-Mixtral-8x7B-DPO | GPT-4 Data, Open datasets |
| MoE v2 | Llama3-8B | LLaMA-MoE-v2-3_8B-2_8-sft | SFT Data |
| MoE4 | Llama2-7B | LLaMA-MoE-v1-3_5B-4_16 | SlimPajama |
| MoE 3B | Llama2-7B | LLaMA-MoE-v1-3_0B-2_16 | SlimPajama |
| MoE2 | Llama2-7B | LLaMA-MoE-v1-3_5B-2_8 | SlimPajama |
| MoE3B-SFT | Llama2-7B | LLaMA-MoE-v1-3_0B-2_16-sft | SlimPajama, SFT Data |
| MoE2-SFT | Llama2-7B | LLaMA-MoE-v1-3_5B-2_8-sft | SlimPajama, SFT Data |
| MoE4-SFT | Llama2-7B | LLaMA-MoE-v1-3_5B-4_16-sft | SlimPajama, SFT Data |
| Qwen1.5 MoE | Qwen-1.8B | Qwen1.5-MoE-A2.7B | Qwen Base Corpus |
| Minicpm MoE | Minicpm-2B | MiniCPM-MoE-8x2B | MiniCPM Data |
| LLaVA | Llama2-7B | llava-v1.5-7b | LAION, GPT instructions, etc. |
| Video | Llama2-7B | Video-LLaVA-7B-hf | Caption, QA |
| VL | Qwen2-7B | Qwen2-VL-7B-Instruct | Image-text, OCR, etc. |
| Audio | Qwen-7B | Qwen-Audio | Speech, Sound, etc. |
| Audio2 | Qwen-7B | Qwen2-Audio-7B | Audio-text, Voice Chat |
| VL | Qwen-7B | Qwen-VL | Image-text, OCR, etc. |
| VL | Qwen2.5-7B | Qwen2.5-VL-7B-Instruct | Visual recognition, Document parsing, etc. |
| VL | Qwen2.5-3B | Qwen2.5-VL-3B-Instruct | Visual recognition, Document parsing, etc. |
| Next | Llama3-8B | llama3-llava-next-8b-hf | LLaVA-NeXT Data |
| LLaVA | Llama2-13B | llava-v1.5-13b | LAION, GPT instructions, etc. |
| RLHF | Open-llama3B | hh_rlhf_rm_open_llama_3b | Anthropic HH-RLHF |
| Reason | Qwen2.5-7B | Nemotron-Research-Reasoning-Qwen-1.5B | Math, Code, etc. |
| Zero | Qwen2.5-1.5B | Open-Reasoner-Zero-1.5B | AIME 2024, MATH500, etc. |
| DPO | Mixtral | Nous-Hermes-2-Mixtral-8x7B-DPO | GPT-4 Data, Preference pairs |
| DPO | Mistral-7B | Nous-Hermes-2-Mistral-7B-DPO | GPT-4 Data, Preference pairs |

**Table 5 – continued from previous page**

| Abbreviation | Base Model | Full Model Name | Train Corpus |
|---|---|---|---|
| Dolphin | Mistral-7B | dolphin-2.6-mistral-7b-dpo | UltraFeedback, Magicoder, etc. |
| DPO | Minicpm-2B | MiniCPM-2B-dpo-bf16 | ShareGPT, UltraChat, etc. |
| GRPO | Qwen3-4B | Qwen3_Medical_GRPO | Medical dataset |
| RLHF | Chatglm-6B | chatglm-fitness-RLHF | SFT, Reward Model, etc. |
| DPO | Llama3-8B | LLaMA3-iterative-DPO-final | UltraFeedback, Preference sets |
| Minitron-Depth | Llama-3-8B | Llama-3.1-Minitron-4B-Depth-Base | Nemotron-4 15B corpus |
| Minitron-Width | Llama-3-8B | Llama-3.1-Minitron-4B-Width-Base | Nemotron-4 15B corpus |
| Sheared 2.7B-P | Llama2-7B | Sheared-LLaMA-2.7B-Pruned | RedPajama |
| Sheared 2.7B-S | Llama2-7B | Sheared-LLaMA-2.7B-ShareGPT | ShareGPT |
| Sheared 2.7B | Llama2-7B | Sheared-LLaMA-2.7B | RedPajama |
| Sheared 1.3B-P | Llama2-7B | Sheared-LLaMA-1.3B-Pruned | RedPajama |
| Sheared 1.3B | Llama2-7B | Sheared-LLaMA-1.3B | RedPajama |
| Sheared 1.3B-S | Llama2-7B | Sheared-LLaMA-1.3B-ShareGPT | ShareGPT |
| Llama3-1B | Llama3-8B | Llama-3.2-1B | Llama3-8B logits, Safety data |
| Llama3-3B | Llama3-8B | Llama-3.2-3B | Llama3-8B logits, Safety data |

## H  IMPLEMENTATION DETAILS

We employ the Linear Kernel ($k(X, Y) = XY^\top$) for Centered Kernel Alignment (CKA) due to its computational efficiency. To mitigate the finite-sample bias inherent in standard HSIC estimations, we utilize the Unbiased CKA (UCKA) estimator. As for module selection, our method operates on two specific sets of weights: first, we utilize the intersection of the word embeddings ($W_{emb}$) to solve the Linear Assignment Problem (LAP), allowing us to accurately recover the permutation ($P$) and signature ($D$) matrices; second, we compute the final fingerprinting scores using the Query ($W_Q$) and Key ($W_K$) weights, as their transformations are strictly constrained.

To address structural discrepancies such as differing layer counts, we identify the optimal layer correspondence by maximizing the total similarity; specifically, we solve the assignment problem on a cost matrix constructed from the pairwise UCKA scores of $W_Q$ and $W_K$ between all source and target layers.

## I  EMPIRICAL VALIDATION OF ROBUSTNESS AGAINST WEIGHT MANIPULATIONS

In Section 4, we theoretically analyze potential weight manipulations, including constant scaling, signature matrix multiplication, permutations, and orthogonal transformations. We now provide empirical evidence to support the analysis. Specifically, we apply these manipulations to five representative models, Llama-2-7B, Qwen2-7B, Mistral-7B, Llama-3-8B, and Gemma-7B, and evaluate AWM's robustness under each setting.

As shown in Table 6, AWM achieves a 100% detection rate across all tested model–manipulation pairs. These results not only align with our theoretical derivations, but also validate the design of AWM, including LAP and UCKA.

Table 6: AWM-detected similarity scores under the weight manipulations in Section 4.

| Manipulation / Model | Llama-2-7B | Qwen2-7B | Mistral-7B | Llama-3-8B | Gemma-7B |
|---|---|---|---|---|---|
| Permutation | 100% | 100% | 100% | 100% | 100% |
| Signature | 100% | 100% | 100% | 100% | 100% |
| Constant Scaling | 100% | 100% | 100% | 100% | 100% |
| Orthogonal Trans. | 100% | 100% | 100% | 100% | 100% |

