# OpenReview forum: "AWM: Accurate Weight-Matrix Fingerprint for Large Language Models"
_ICLR.cc/2026/Conference — ICLR 2026 Poster_

### Official Review · Reviewer_Wyog · 2025-10-29

**Soundness:** 4
**Presentation:** 4
**Contribution:** 3
**Rating:** 6
**Confidence:** 4

**Summary:**

The authors propose AWM, a method based on centered kernel alignment to fingerprint and compare LLMs. It is more accurate than competing methods such as HuRef and REEF on a large set of different LLMs of different families. However, it requires knowledge of all the weights of the LLMs to be compared.

**Strengths:**

- Empirically the new proposed algorithm is quite strong, improving upon previous methods such as HuRef and REEF in distinguishing between related and unrelated LLMs on a fairly large set of LLMs tested.

- The algorithm works directly with weight matrices and only uses centered kernel alignment computation and the Hungarian algorithm for finding permutations, and is therefore very efficient.

- The authors give a detailed list of potential weight manipulations on different parts of the transformer architecture by an attacker in Section 4 and explain how this guides the design of their algorithm.

**Weaknesses:**

- This fingerprinting method requires access to the weights of the models to be compared, which isn't always possible unless both models are open-sourced. This limits the applicability of the method compared to other fingerprinting approaches.

- In Section 4 the authors discuss how an attacker might manipulate the weights to evade detection. However, these manipulations are never evaluated against the algorithm in the experiments. This creates a gap in the argument of the paper.

**Questions:**

- Have the authors consider how to apply their method to other forms of manipulations/infringement such as model distillation?

---

> ### Author Response · Authors · 2025-11-23
>
> We thank the reviewer for the positive assessment of AWM, including its strong empirical performance, efficiency, and the comprehensive analysis of manipulations. In what follows, we respond to each concern raised by the reviewer.
>
> > **[W1&Q1]** AWM requires direct access to model weights, limiting its applicability. Can AWM be extended to handle other forms of misuse such as model distillation?
>
> We are grateful to the reviewer for this clarifying question. As discussed in Section 4, AWM is explicitly designed for the white-box setting like REEF and HuRef, where one suspects that a model directly or indirectly inherits weights from a base model, and in this regime weight-matrix similarity is the right signal to examine. By contrast, pure API-based knowledge distillation can produce students that mimic the teacher’s behaviour while having almost uncorrelated weights, so we view these cases as primarily the domain of black-box fingerprinting methods. That said, the two approaches are complementary: lightweight black-box probes can be used to broadly flag potential distillation or copying cases, and, whenever model weights later become available (e.g., for an open model or in an audit), AWM can serve as a high-precision white-box check on weight inheritance. In particular, AWM’s strict control of false positives makes it well suited for third-party verification settings, where regulators, courts, or other independent auditors confidentially examine model weights to assess potential infringement.
>
> > **[W2]** Manipulations in Section 4 are not tested against AWM.
>
> We agree that explicitly evaluating the discussed manipulations is important for closing the gap in our argument. In addition to the analysis in Section 4, we now empirically test the four manipulation strategies considered in Section 4.5, including permutations, signature matrix multiplications, constant scaling, and orthogonal transformations. We have included these detailed results in **Appendix I** of the revised paper. The AWM-detected similarity rates are also reported below:
> |Attack\Model|Llama-2-7B|Qwen2-7B|Mistral-7B|Llama-3-8B|Gemma-7B|
> |-|-|-|-|-|-|
> |Permutation|100%|100%|100%|100%|100%|
> |Signature|100%|100%|100%|100%|100%|
> |Constant Scaling|100%|100%|100%|100%|100%|
> |Orthogonal Transformations|100%|100%|100%|100%|100%|
>
> As can be seen above, AWM successfully detects the manipulations discussed in Section 4. These results support the arguments of the paper.

---

### Official Review · Reviewer_z54L · 2025-10-30

**Soundness:** 3
**Presentation:** 3
**Contribution:** 3
**Rating:** 6
**Confidence:** 3

**Summary:**

This paper proposes AWM, a training-free, white-box fingerprint for LLMs that combines LAP-based layer matching with an unbiased CKA similarity to measure model lineage directly from weights. The experimental suite spans six common post-training transformations (SFT, continued pretraining, RL-based alignment, multimodal extension, pruning, and MoE upcycling) over 60 positive and 90 negative pairs. The method is computationally efficient (single-GPU, sub-minute per pair) and requires no instrumenting or watermarking of the target model.

**Strengths:**

1.The method is minimalist and targeted, focusing on embeddings and Q/K, which makes LAP+linear CKA implementation straightforward and efficient (single-GPU, sub-minute).
2.Dynamic layer assignment via LAP addresses the weakness of fixed layer-to-layer comparisons and tolerates moderate architectural edits.
3.It requires no data collection, retraining, or watermark insertion, so it does not degrade model quality and is easy to operationalize.
4.Empirically, the approach cleanly separates derived from independent models (AUC=1.0, TPR@1%FPR=100%), implying near-zero false-positive risk.

**Weaknesses:**

1.The pipeline relies on overlaps in token embeddings and Q/K weights; if the suspect model re-trains the tokenizer or heavily replaces early blocks, the LAP matching can become unstable and |Z| may drop.
2.In cases of knowledge distillation, AWM will likely fail because cross-model weights exhibit near-zero statistical correlation despite functional similarity. Notably, recent alleged infringement incidents (e.g., API-based distillation claims) fall into this category.
3.Although LAP provides some structural elasticity, it presupposes roughly comparable sets of alignable matrices. If the suspect model adds/removes many layers or alters layer roles, matching can break or produce erroneous alignments. While permutation invariance should not affect CKA in principle, aggressive layer reordering can still mislead the LAP stage and degrade end-to-end performance.
4.Since the method is simple and single, the attacker can try to perform some lossless transformation on the AWM model to reduce the CKA similarity but maintain the model output. This is not difficult to achieve.
5.The approach evaluates one-to-one similarity and does not address models fused from multiple sources (e.g., model 𝐶 partly derived from 𝐴 and 𝐵).
6.The only comparison methods are HuRef and REEF. Why are there no comparisons with baselines used in REEF, such as PCS and ICS?
7.The related work section recommends adding some black box-based methods. Currently, piracy through APIs is also very common.

**Questions:**

1.The first step of the method relies on a shared vocabulary. How should we handle cases where the suspect model retrains or replaces its tokenizer/vocabulary?
2.Does the author have any insights on piracy cases involving knowledge distillation? Could lightweight black-box probes be combined with AWM to first flag potential distillation cases?
3.When the architecture undergoes significant depth/width changes, cross-layer sharing, or reordering of blocks, can you constrain the LAP with structural priors to prevent misalignment?
4.Since the method is very simple, it becomes easier to attack. How can this issue be addressed?
5.Have you considered approaches for mixed-origin cases, for example, solving multiple partial LAPs and reporting a mixed (or mixture) score?
6.Release an evaluation sheet (model names/versions, checkpoints, tokenizer specs, training corpus tags), and precise CKA/LAP implementation details (module selection, sampling of parameter blocks, normalization, kernel choices).
7.It is recommended to add some comparison methods and, if possible, add explicit ablation experiments

---

> ### Author Response · Authors · 2025-11-23
>
> We thank the reviewer for the thoughtful and positive assessment of our work, including AWM's computational efficiency, robustness, zero influence on model performance and near-zero false positive rates. Below, we address each of the reviewer’s points one by one.
>
> > **[W1&Q1]** AWM relies on a shared vocabulary and token embedding overlaps. Thus, it is vulnerable to the cases where the early blocks of the model is replaced, or the tokenizer is replaced or retrained.
>
> We respectfully believe that AWM is more robust than this concern suggests, as detailed below.
> - First, although one can in principle heavily modify or replace the **early layers** of an LLM, **prior work [1] shows that such interventions typically cause large performance drops that are hard to recover even with further training**. This conflicts with the usual goal of reusing a strong base model.
>
> - Second, **AWM is applicable to the case where the tokenizer is replaced or retrained**. By noticing the public discussion around some certain MoE model and Qwen-2.5-14B, which only share roughly 21% of vocabulary tokens, we apply AWM to detect similarity of the two, and report an absolute Z-score of 248.48. This result suggests that even though the tokenizer is heavily replaced or retrained, AWM can remain effective.
>
> - Last, **AWM does not rely heavily on the overlap of tokens**. We run experiments for all scenarios in Table 2 with varying numbers of shared tokens, and report the average absolute Z-score for each scenario in Appendix F.1. The results suggest that AWM remain effective and maintain high average absolute Z-scores (>100) in most cases even if there are very limited overlapping tokens (~100 tokens) in the vocabulary. We summarize the results in the following table.
>
> |Modification\Overlapping Tokens|10|100|500|1000|3000|5000|10000|
> |-|-|-|-|-|-|-|-|
> |SFT|354.32|354.32|354.32|354.32|354.32|354.32|354.32|
> |Continual Pretrain|5.40|147.64|192.08|204.63|208.02|208.21|210.89|
> |Upcycling|44.22|248.28|267.94|275.18|281.71|286.05|287.85|
> |Multi modal|205.50|332.80|334.52|334.81|334.82|334.82|334.78|
> |RL|355.79|355.79|355.79|355.79|355.79|355.79|355.79|
> |Pruning|76.46|250.94|255.51|255.87|255.93|256.06|257.72|
>
> Reference
> [1] Huang, Hanbo, et al. "Archilles' Heel in Semi-open LLMs: Hiding Bottom against Recovery Attacks." (2024).
>
>
> > **[W2&Q2]** AWM may fail to detect API-based distillation. Any insights on piracy cases involving knowledge distillation? Could lightweight black-box probes be combined with AWM to first flag potential distillation cases?
>
> We appreciate the reviewer’s careful consideration of our work.
> - First, **AWM is explicitly designed for the white-box setting where one suspects that a model directly or indirectly inherits weights from a base model** (see Section 4.1). In this regime, weight-matrix similarity is the right signal to examine, and our experiments show that AWM can offer high confidence with a very low false-positive rate. In contrast, in pure API-based knowledge distillation the student can mimic the teacher’s behaviour while having almost uncorrelated weights, so weight-based fingerprint, including ours, has limited power. We therefore view such API-only distillation cases as primarily the domain of black-box fingerprinting methods, rather than the main target of AWM.
>
> - Second, **black-box and white-box approaches are complementary rather than competing**.  Since AWM and black-box methods detect matrix weight similarity and output similarity respectively, combining the two approaches can yield a conclusion on the independency of both the model weights and the data used in training.
>
> - Finally, **AWM’s accuracy and strict control of false positives make it well suited for third-party verification scenarios**, where model owners, regulators, or courts can confidentially submit model weights to an independent auditor. In such settings, AWM can provide strong evidence about whether one model’s weights are independently trained or derived from another, naturally complementing black-box tests that focus on behavioural similarity and potential API-based distillation.

---

> > ### Author Response · Authors · 2025-11-23
> >
> > > **[W3&Q3]** Will AWM work when the architecture undergoes significant depth/width changes, layer removal, layer addition, cross-layer sharing, or reordering of blocks?
> >
> > We show that AWM remains effective under these architectural modifications, as detailed below.
> > - First, **layer removal/layer addition/layer reordering/cross-layer sharing can be effectively detected by AWM** since Algorithm 1 can find an optimal match of layers by performing LAP on layer-wise similarities.  Empirically, we find that
> >     - The detection of **layer removal** is effective. In the "pruning" part of Table 2, models including Sheared-LLaMA have gone through layer removal, but still presents a high similarity to the base models under AWM detection (most showing absolute Z-scores higher than 200).
> >     - AWM successfully flags **layer addition**.  We additionally collect 3 model pairs within this case. Moreover, we also train an additional LLaMA-3.2-3B-two-layer by adding two radomly initialized layers to LLaMA-3.2-3B and training on the RedPajama dataset to support this viewpoint. In all the four cases AWM remains effective as well, and the results are summarized in the table below.
> >     - For **layer reordering** and **cross-layer sharing**, **the case is similar theoretically**. Though we do not find examples of such cases, LAP over layer-wise similarities in Algorithm 1 can figure out layer pairs as layer removal and layer addition since the computation of layer-wise similarities is identical.
> >
> > |Modification|Model Pair|Absolute Z-Score|
> > |-|-|-|
> > |Add layers|Mistral-7B-v0.1 vs SOLAR-10.7B-v1.0|323.5893|
> > | |Yi-6B vs Yi-9B|329.5536|
> > | |Llama2-7b vs LLaMA-Pro-8B|355.8036|
> > | |LLaMA-3.2-3B vs LLaMA-3.2-3B-two-layers|355.1029|
> >
> >
> > - Second, **AWM still works in cases where the width of the layers changes**. As discussed in  "Robust Recovery of Weight Similarities" in Section 5, width changes, especially pruning, can be viewed as a semi-orthogonal transformation which can be tolerated by UCKA to some extent. The results in the "Pruning" part of Table 2 further validate this point: though many pairs (e.g. LLaMA3-8B vs LLaMA3-3B) have gone through significant width pruning, AWM still successfully flags the similarity.
> >
> > > **[W4&Q4]** AWM is very simple and is relatively easy to attack.
> >
> > The simple design of AWM requires minimum additional parameters, from which AWM's robustness benefits (see "Robust Recovery of Weight Similarities" in Section 5). Compared to the baselines (HuRef, REEF, Intrinsic Fingerprint), **our approach is the state-of-the-art in terms of robustness, a result of the simple yet effective design**. Moreover, AWM is built upon a systematic analysis of matrix weight manipulations in Section 4. **A variety of attacks, including signature matrices, permutations, constant scaling and orthogonal transformations (see Section 4.5), are fully detectable with AWM**  due to the properties of UCKA (see "Central Kernel Alignment (CKA)" in Section 3) and LAP reconstruction of the permutation attacks. Moreover, the LAP matching of layers in Algorithm 1 also prevents layer-wise attacks, as discussed in previous responses. Hence, AWM is a simple detection method but robust to various forms of attacks.
> >
> > > **[W5&Q5]** Will AWM work if one model is fused from multiple sources?
> >
> > We agree that mixed-origin and merged models are an important use case. Our current formulation of AWM is pairwise and evaluates similarity between one suspect model and one candidate source, but this does not preclude handling multi-source merges. In fact, for publicly documented merged models such as mergekit-community/Qwen3-1.5B-Instruct (a TIES merge of several Qwen2.5-1.5B variants) and SakanaAI/EvoLLM-JP-v1-7B (an evolutionary merge of multiple 7B models), we apply AWM separately between the merged model and each of its published base models. **We find that AWM assigns consistently high similarity scores to the true source models, allowing us to recover which components were used in the merge**. Results are shown in the following table.
> >
> > |Offspring Model|mergekit-community/Qwen3-1.5B-Instruct|||SakanaAI/EvoLLM-JP-v1-7B|||
> > |-|-|-|-|-|-|-|
> > |Source Model|Qwen2.5-1.5B-Instruct|Qwen2.5-Math-1.5B-Instruct|Qwen2.5-Coder-1.5B-Instruct|shisa-gamma-7b-v1|WizardMath-7B-V1.1|Abel-7B-002|
> > |Absolute Z-Score|84.2679|274.4107|144.6964|353.0179|351.5536|351.5179|

---

> ### Author Response · Authors · 2025-11-23
>
> > **[W6&Q7]** Why no baselines like PCS and ICS? Add comparison methods and explicit ablation experiments.
>
> PCS and ICS are both proposed in HuRef, where ICS is more robust than PCS as suggested in HuRef. Therefore, we choose ICS as HuRef's baseline. Besides, we add Intrinsic Fingerprint [1] and PCS as additional baselines in Table 3, which are still outperformed by AWM. We also present them in the following table along with the performance of our method:
>
> |Method|Metric|SFT|CPT|UP|MM|RL|PR|All|
> |-|-|-|-|-|-|-|-|-|
> |Intrinsic Fingerprint|\|Z\| ↑|1.535|1.193|1.408|1.532|1.542|1.141|1.392|
> |Intrinsic Fingerprint|AUC ↑|1.000|0.896|0.969|1.000|1.000|0.876|0.957|
> |Intrinsic Fingerprint|pAUC ↑|1.000|0.422|0.800|1.000|1.000|0.400|0.770|
> |Intrinsic Fingerprint|TPR@1%FPR ↑|1.000|0.300|0.800|1.000|1.000|0.400|0.750|
> |PCS|\|Z\| ↑|73.786|74.650|0.727|14.950|163.399|17.226|57.457|
> |PCS|AUC ↑|0.958|0.959|0.791|0.802|0.984|0.666|0.860|
> |PCS|pAUC ↑|0.656|0.600|0.078|0.500|0.900|0.100|0.472|
> |PCS|TPR@1%FPR ↑|0.500|0.600|0.000|0.500|0.900|0.100|0.433|
> |Ours|\|Z\| ↑|353.788|219.155|287.634|334.556|355.250|267.233|302.936|
> |Ours|AUC ↑|1.000|1.000|1.000|1.000|1.000|1.000|1.000|
> |Ours|pAUC ↑|1.000|1.000|1.000|1.000|1.000|1.000|1.000|
> |Ours|TPR@1%FPR ↑|1.000|1.000|1.000|1.000|1.000|1.000|1.000|
>
>
> Moreover, we conduct an ablation study on  mechanisms of AWM, including CKA kernel selection (linear/rbf) and the unbiased/biased design of CKA, and summarize the average absolute Z-scores in the scenarios of Table 2 in Appendix F.2. We also present these results here:
>
> |Modification\CKA Design|Linear (Unbiased)|Linear (Biased)|RBF (Unbiased)|RBF (Biased)|
> |-|-|-|-|-|
> |SFT|356.0223|18.3099|527.3949|11.5843|
> |Continual Pretrain|217.5003|11.2954|320.9173|7.0701|
> |Upcycling|291.6191|15.1994|432.1476|9.6776|
> |Multi Modal|336.6757|17.3766|498.8918|11.0056|
> |RL|357.5001|18.3850|529.6448|11.6330|
> |Pruning|268.9175|13.8985|394.8391|8.7331|
>
> In addition, we perform an ablation study on the number of overlapped tokens used, and the results are shown in Appendix F.1 and the response to W1&Q1.  The results suggest that AWM can effectively detect model similarities even if the number of shared vocabulary tokens is extremely low (~100 tokens).
>
> References
> [1] Yoon, Do-hyeon, et al. "Intrinsic Fingerprint of LLMs: Continue Training is NOT All You Need to Steal A Model!." arXiv preprint arXiv:2507.03014 (2025).
>
> > **[W7]** Add discussions on black-box methods in Related Works.
>
> Thank you for pointing this out! We've revised the paper accordingly and added the discussion of black-box approaches in Related Works.
>
> > **[Q6]** Release an evaluation sheet (model names/versions, checkpoints, tokenizer specs, training corpus tags), and precise CKA/LAP implementation details (module selection, sampling of parameter blocks, normalization, kernel choices).
>
> - We have added a comprehensive evaluation sheet in Appendix G (Table 5). This table details the 60 offspring models used in our study, mapping each abbreviation to its full model name, base model, and relevant training corpus. To facilitate reproducibility, every entry in the “Full Model Name” column is hyperlinked directly to the corresponding official model checkpoint on the Hugging Face Hub.
> - As for implementation details, we employ the Linear Kernel ($k(X, Y) = XY^\top$) for Centered Kernel Alignment (CKA) due to its simplicity and computational efficiency. To mitigate the finite-sample bias inherent in standard HSIC estimations, we utilize the Unbiased CKA (UCKA) estimator. As for module selection, our method operates on two specific sets of weights: first, we utilize the intersection of the word embeddings ($W_{emb}$) to solve the Linear Assignment Problem (LAP), allowing us to accurately recover the permutation ($P$) and signature ($D$) matrices; second, we compute the final fingerprinting scores using the Query ($W_Q$) and Key ($W_K$) weights, as their transformations are strictly constrained. To address structural discrepancies such as differing layer counts, we identify the optimal layer correspondence by maximizing the total similarity; specifically, we solve the assignment problem on a cost matrix constructed from the pairwise UCKA scores of $W_Q$ and $W_K$ between all source and target layers.

---

> > ### Comment · Reviewer_z54L · 2025-11-26
> >
> > Thanks to the authors' efforts, it addresses my major concerns. Another reviewer has promised to raise the score, and I will adjust (raise) my rating, referring to other reviewers' comments and the reference value of this research to the entire community.

---

### Official Review · Reviewer_fYwn · 2025-10-30

**Soundness:** 3
**Presentation:** 2
**Contribution:** 3
**Rating:** 4
**Confidence:** 3

**Summary:**

This paper presents a method for LLM intellectual property (IP) protection, aiming to overcome the vulnerabilities of current fingerprinting schemes, which are often not robust against intensive post-training and malicious weight manipulations. The authors introduce AWM (Accurate Weight-Matrix Fingerprint), a novel, training-free, intrinsic weight-based fingerprinting scheme designed to be highly robust to such modifications. The method's core mechanism leverages the Linear Assignment Problem (LAP) and an unbiased Centered Kernel Alignment (CKA) similarity metric. This process first extracts permutation and signature transformations from the word embedding matrices via LAP , and then uses the CKA-based metric to robustly compare the Q and K matrices across layers, a design intended to neutralize the effects of parameter manipulations like orthogonal transformations. Experimental results demonstrate that AWM achieves perfect classification scores and shows exceptional robustness.

**Strengths:**

- The idea of this paper is novel.
- This paper presents promising results.
- This paper provides a comprehensive theoretical analysis.

**Weaknesses:**

- Dependency on Word Embeddings: The method relies heavily on the word embedding matrix. An attacker could potentially defeat the detection by freezing all other layers and only replacing or retraining the embedding layer, which may cause the method to fail.

- Limited Detection Scope: The method's scope is restricted to $W_Q$ and $W_K$ matrices. An attacker could steal other components (e.g., implanting stolen FFN blocks into an MoE model). This form of partial theft would likely go undetected.

- Other suggestions for improving the writing:
  - The theoretical analysis in Section 4 is somewhat confusing.  It may be better for the authors to first introduce the basic ideas and conclusions at the beginning of the section to improve readability.

  - The Related Work section on fingerprinting is limited, focusing mostly on traditional classification models and only two LLM fingerprinting papers. It may be better for the authors to conduct a more exhaustive survey to enrich this section.

**Questions:**

Please refer to the Weaknesses section.

---

> ### Author Response · Authors · 2025-11-23
>
> We thank the reviewer for the careful reading of our work and for highlighting its strengths, including novelty, performance and theoretical analysis. In what follows, we will respond to your concerns and revise the paper accordingly.
>
> > **[W1]** Dependency on Word Embeddings: retrianing or replacing the embedding layer with other layers freezed could potentially defeat AWM.
>
> We appreciate the reviewer’s thoughtful comment regarding this issue.
> - First, **retraining the embedding layer from scratch may degrade the model's performance significantly**, which may contradict with the purpose to reuse the model's performance. We conduct an experiment on Llama3-3B, where the embedding layer is re-initialized and trained for on the RedPajama dataset with other layers freezed. We observe a significant drop in the model's performance on popular benchmarks, arguing the validity of such a manipulation approach. The performance comparison is presented in the following table:
>
> | Model\Benchmark | LAMBADA (OpenAI version) | ARC-E | LAMBADA (Standard version) | ARC-C | WinoGrande | PIQA | HellaSwag | SciQ  | RACE | Average |
> |-|-|-|-|-|-|-|-|-|-|-|
> |Re-initialized embedding, trained on the RedPajama with other layers freezed|3.4|29.9|2.8|20.2 |53.9|55.6|26.2|47.5|23.7|29.2|
> |LLaMA3-3B|70.1|74.5|63.7|42.2|69.0|76.8|55.4|95.5|39.4|65.2|
>
> - Second, **replacement of embeddings and the tokenizer can be detected by AWM**. We study a real-world case where the tokenizer may be replaced and retrained. Noticing the heated debate on the similarity between some certain MoE model and Qwen-2.5-14B, we apply AWM to this case. Even though the tokenizers of the two models differ substantially (roughly 78% of their vocabulary tokens are not shared), AWM reports a Z-score of 248.48, flagging the similarity. In other words, heavy replacement of tokenizers and embeddings does not remove the deeper structural alignment that AWM captures in the remaining layers in this case, and the method continues to flag this pair as highly similar.
>
> - Finally, **AWM does not rely heavily on the overlap of vocabulary tokens and related word embeddings**. To test how much AWM depends on overlapping tokens, we run an ablation over the number of shared tokens and report the results in Appendix F.1. Within such cases, AWM assigns high Z-scores and detects similarity (average absolute Z-score over 100) in most cases even if the number of overlapping tokens is low (~100 tokens), indicating that it does not heavily rely on a large set of aligned word embeddings. We also present the results in the following:
>
> |Modification\Overlapping Tokens|10|100|500|1000|3000|5000|10000|
> |-|-|-|-|-|-|-|-|
> |SFT|354.32|354.32|354.32|354.32|354.32|354.32|354.32|
> |Continual Pretrain|5.40|147.64|192.08|204.63|208.02|208.21|210.89|
> |Upcycling|44.22|248.28|267.94|275.18|281.71|286.05|287.85|
> |Multi modal|205.50|332.80|334.52|334.81|334.82|334.82|334.78|
> |RL|355.79|355.79|355.79|355.79|355.79|355.79|355.79|
> |Pruning|76.46|250.94|255.51|255.87|255.93|256.06|257.72|

---

> ### Author Response · Authors · 2025-11-23
>
> > **[W2]** Limited Detection Scope: other components (e.g., implanting stolen FFN blocks into an MoE model) are unlikely to be detected by AWM.
>
> - First, **AWM‘s effectiveness can extend to other components of a model**, even though it is designed to detect the similarity of Q,K matrices. In the following table we show the average Z-score of FFN matrices, i.e. Gate, Up and Down matrices, where AWM still flags significant similarity.  Even if the offspring model has gone through extensive continual pretraining (e.g., 5.5T tokens for Qwen2.5-Coder-7B), AWM still successfully detects the similarity.
>
> |Pair|Average Absolute Z-score|
> |-|-|
> |Llama-2-7B vs Llemma-7B|194.9922|
> |Llama-2-7B vs CodeLlama-7B-hf|200.7813|
> |Llama-2-7B vs CodeLlama-7B-Python-hf|196.7552|
> |Gemma-2B vs CodeGemma-2B|184.7341|
> |Gemma-7B vs CodeGemma-7B|256.0771|
> |Qwen2.5-7B vs Qwen2.5-Math-7B|190.4809|
> |Qwen2-7B vs Qwen2.5-Coder-7B|106.3877|
> |Qwen2-7B vs Qwen2-Math-7B|198.5877|
> |Llama-2-70B vs CodeLlama-70B-hf|191.6779|
> |Llama-2-70B vs CodeLlama-70B-Python-hf|183.4636|
>
> - Apart from these results, we notice that there's a heated debate on some certain MoE model recently, whose FFN blocks may originate from Qwen-2.5-14B. Hence, we test the similarity between that model's FFN weight matrices and Qwen-2.5-14B's using AWM, and report the similarity scores as follows:
>
> |Component|Dynamic 64 mlp experts, dim=[1344,5120]|||Shared 4 mlp expert dim=[5376,5120]|||
> |-|-|-|-|-|-|-|
> |**Block**|**Down**|**Up**|**Gate**|**Down**|**Up**|**Gate**|
> |Absolute Z-Score|75.20|71.45|121.13|191.38|161.16|222.45|
>
> > **[W3]** Section 4 is confusing.
>
> Thanks for your helpful comments! We've revised the paper accordingly, adding the basic ideas and conclusions at the beginning of Section 4.  Apart from that, we also modify the writing of Section 4 to improve the paper's readability.
>
> > **[W4]** The Related Work section on fingerprinting is limited,
>
> Thank you for your helpful feedback! We've enriched the Related Work section, adding more discussions on white-box fingerprint approaches and black-box ones. Please refer to "Related Work" in the revised paper.

---

> ### Comment · Reviewer_fYwn · 2025-11-24
>
> Thank you for the rebuttal, and it addresses most of my concerns. I will raise my rating.

---

### Official Review · Reviewer_tHGT · 2025-10-31

**Soundness:** 3
**Presentation:** 4
**Contribution:** 3
**Rating:** 8
**Confidence:** 3

**Summary:**

This paper presents a mechanism to "fingerprint" an LLM. This is to find out if an LLM is trained from scratch or is a finetuned/derived version of a different LLM. The paper evaluates their methods on multiple models, different post training techniques and they achieve perfect scores in all classification metrics.

**Strengths:**

The computation completes within 30 seconds on a single NVIDIA 3090 GPU.

The method does not require any additional training and does not impact the LLM's performance (as other watermarking approaches)

The evaluation is very comprehensive across multiple llama models and multiple offspring variations where models are post trained with sft, continued pretraining, RL.

They also present a false positive evaluation with 90 known to be unrelated pairs, where they show other approach like REEF shows significant false positives mostly for models that use same training data sources.

**Weaknesses:**

Maybe lack of experiments with larger models and other model architectures like MoE.

**Questions:**

Any thoughts on how this will work with MoE models?

Any thoughts on how this will work if the post training is done with LoRA methods and or adding new layers of randomly initialized parameters.

**Details Of Ethics Concerns:**

Perhaps the results of privacy and IP concerns for some of the model pairs should be  look up thoroughly, even if the paper claimed them to be as false positives from other methods (see Fig 2).

---

> ### Author Response · Authors · 2025-11-23
>
> Thank you for your positive assessment of the computational efficiency, training-free design, comprehensive multi-model evaluation, and extensive false-positive analysis in the paper. We appreciate your time and constructive feedback. Below, we provide point-by-point responses to your comments.
> > **[W1&Q1]** Will AWM work with larger models and other model architectures, such as MoE?
>
> AWM remains effective in these cases.
> - For larger models, we show in the "Continual Pretrain" part of Table 2 that AWM successfully detects the similarity among Llama2-70B and its continual-pretrained offsprings, CodeLlama-70b-hf and CodeLlama-70b-Python-hf. In both of the two cases,  AWM's absolute Z-scores are over 230, flagging significant similarity.
> - For MoE models, we conduct two lines of tests, dense base models vs their upcycled MoE offsprings,  and MoE base models vs their offsprings.
>   - For the first aspect, we show in the "Upcycling" part of Table 2 that AWM still works. To be specific, we test the similarity of Mistral-7B, Llama3-8B, Llama2-7B, Qwen-1.8B, Minicpm-2B, and their upcycled variants, and find that AWM accurately detects the similarity with absolute Z-scores at least 10 times larger than previous methods.
>   - For the second, we compare Qwen3-235B-A22B-FP8 against its offsprings. In such a scenario, AWM still quickly flags the similarity. The results are shown in the following table.
>
> |Base|Qwen3-235B-A22B|||
> |-|-|-|-|
> |Offspring|Qwen3-235B-A22B-Thinking-2507|Qwen3-VL-235B-A22B-Instruct|Qwen3-235B-A22B-Instruct-2507|
> |Absolute Z-score|349.6607|343.9464|350.0179|
>
> > **[Q2]** Any thoughts on how this will work if the post training is done with LoRA methods and or adding new layers of randomly initialized parameters.
>
> AWM remains applicable in both scenarios.
> - When **new layers** are added to the model, **Algorithm 1 first finds an optimal match between layers of the base model and the derived model by performing LAP on layer-wise similarities**. We then compute UCKA on these matched layer pairs, from which AWM successfully recovers the similarity. Empirically, we collect 3 additional model pairs under this regime, and test AWM against them. We also conduct an experiment where  two randomly initialized layers are added to LLaMA-3.2-3B and the new model is continually pretrained on the RedPajama dataset. We term this model as LLaMA-3.2-3B-two-layers, and use AWM to test its similarity to LLaMA-3.2-3B.  In all of the four cases, AWM successfully identifies the originality of models. We summarize the results in the following table.
>
> Modification|Model Pair|Absolute Z-Score
> |-|-|-|
> Add layers|Mistral-7B-v0.1 vs SOLAR-10.7B-v1.0|323.5893
> | |Yi-6B vs Yi-9B|329.5536
> | |Llama2-7b vs LLaMA-Pro-8B|355.8036
> | |LLaMA-3.2-3B vs LLaMA-3.2-3B-two-layers|355.1029
>
>
> - For **LoRA**, which introduces small low-rank updates to weights, we find empirically that AWM is robust to such perturbations. For example, AWM reports 99.81% similarity between the LoRA-fine-tuned Firefly-LLaMA2-13B and its base model LLaMA2-13B (Table 1), and detects the similarity between ChatGLM-6B and its LoRA-fine-tuned variant with a Z-score of 355.27 (Table 2). We additionally collect 5 pairs of models where Lora is applied , and perform AWM on these cases. The results suggest that AWM successfully flags model similarities. We summarize the results as follows:
>
> Modification|Model Pair|Absolute Z-Score
> |-|-|-|
> LoRA|Llama2-7b vs Llama-2-Medical-Merged-LoRA|354.9107
> | |Llama-7b-hf vs chinese-llama-lora-7b|241.9821
> | |Llama-7b-hf vs alpaca-lora-7b|352.8750
> | |Llama-13b vs chinese-llama-lora-13b|296.6964
> | |Llama-13b vs ziya-llama-13b-medical-lora|354.0179

---

> ### Comment · Reviewer_tHGT · 2025-11-26
>
> I think the authors did a very good job answering my questions with additional experiments including MOE and expanding to new models.

---

### Author Response · Authors · 2025-12-03
**Summary of Rebuttal for the Area Chair**

Dear Area Chair,

Thank you for overseeing the evaluation of our submission. To facilitate your assessment, we first highlight the main strengths identified in the reviews, and then summarize the key concerns together with our responses.

**Key strengths**

- **Effectiveness and robustness.** Reviewers note that AWM achieves extremely low false‑positive rates (AUC = 1.0, TPR@1% FPR = 100%) across diverse training recipes (SFT, RL, multi‑modal post‑training, continual pretraining) and architectural changes (upcycling, pruning). (Reviewers tHGT, z54L, fYwn, Wyog)

- **Training-free design with zero impact on LLM performance.** Reviewers commended AWM for requiring no training or watermark insertion, thus avoiding the performance degradation typical of competing approaches. (Reviewers tHGT, z54L)

- **Comprehensive theoretical analysis.** Reviewers recognize that AWM is well motivated by a thorough theoretical analysis of potential attacks. (Reviewers fYwn, Wyog)

- **Computational efficiency.** Reviewers praise the lightweight design of AWM, where detection takes at most 30s on a single NVIDIA 3090 GPU per model pair. (Reviewers tHGT, z54L, Wyog)

**Main concerns and our responses**

- **Generalization to diverse architectures and setups.**  We show that AWM’s effectiveness extends to MoE models (c.f. response to Reviewer tHGT, W1&Q1) and remains effective in cases like LoRA finetuning (c.f. response to Reviewer tHGT, Q2). We also demonstrate that AWM can flag cases where a model is fused from mixed sources (c.f. response to Reviewer z54L, W5&Q5) and detect FFN similarities in both curated model pairs and a real‑world case. (c.f. response to Reviewer fYwn, W2)

- **Robustness against structural and component alterations.** We show that AWM remains robust under layer addition (c.f. response to Reviewer tHGT, Q2), layer removal, depth/width changes, cross‑layer sharing, and block reordering (c.f. response to Reviewer z54L, W3&Q3). Furthermore, AWM successfully detects lineage even under replacements of embeddings and tokenizers in real‑world cases, overcoming low vocabulary overlap. (c.f. response to Reviewer fYwn, W1)

- **Robustness against LLM weight manipulations.** We empirically validate AWM against a variety of manipulations discussed in the paper, including constant scaling, signature matrix multiplications, permutations, and orthogonal transforms. (c.f. response to Reviewer Wyog, W2)

- **More baselines and ablation studies.** We include PCS and Intrinsic Fingerprint as additional baselines in our experiments, and AWM still significantly outperforms both. We also conduct ablations on the design of CKA in AWM, covering the selection of kernels and biasedness (c.f. response to Reviewer z54L, W6&Q7). We further show that the effectiveness of AWM does not heavily rely on the number of shared vocabulary tokens. (c.f. response to Reviewer fYwn, W1)

During the discussion phase, **Reviewer fYwn raised the score from 4 to 6 (24/11/2025), and Reviewer z54L promised to raise the score (26/11/2025). Reviewer tHGT also provided positive feedback on 26/11/2025.**

We sincerely appreciate your time and effort in evaluating our submission, and we are glad to provide any further clarification if needed.

---

### Meta-Review · Area_Chair_AGhi · 2026-01-07

**Summary:**

This paper proposes a training-free method for identifying whether a suspect LLM is derived from an existing base model or trained independently. The reviewers appreciated the method's strong empirical performance and computational efficiency. Besides, the authors provided comprehensive theoretical analysis of potential weight manipulations.

**Reviewer Concerns:**

Addressed concerns:
1. Generalization to MoE. The authors provided additional experiments demonstrating AWM's effectiveness on MoE architectures. The results show successful detection. This concern is addressed.

2. Additional baselines and ablations. The authors added PCS and Intrinsic Fingerprint as baselines, and conducted ablation studies on CKA kernel selection. This concern is addressed.

Outstanding concerns:
The method inherently cannot detect API-based knowledge distillation where student weights are uncorrelated with teacher weights. The authors acknowledge this is a limitation of white-box approaches.

**Reviewer Scores:**

Reviewer fYwn mentioned raised their score after the rebuttal addressed concerns about embedding dependency and detection scope. Reviewer z54L acknowledged the rebuttal addressed their concerns about structural modifications, and mixed-origin cases, and would raise the score.

---

### Decision · Program_Chairs · 2026-01-26

Accept (Poster)